# Dynamics of mesoscale brain network during visual discrimination learning revealed by chronic, large-scale single-unit recording

**Tian-Yi Wang[1†], Chengcong Feng[1,2†], Chengyao Wang[1], Chi Ren[1*], Zhengtuo Zhao[1,2*]**

[1]Institute of Neuroscience, Center for Excellence in Brain Science and Intelligence Technology, Chinese Academy of Sciences, Shanghai, China; [2]University of Chinese Academy of Sciences, Beijing, China

## eLife Assessment

This study presents experiments suggesting intriguing mesoscale reorganization of functional connectivity across distributed cortical and subcortical circuits during learning. The approach is technically impressive, and the results are potentially of **valuable** significance. The authors have also made a clear effort to address concerns in revision. However, the strength of evidence remains **incomplete**. Acquisition of data from additional animals in the primary experiment could bolster these findings.

**Abstract** Associating unfamiliar stimuli with appropriate behavior through experience is crucial for survival. While task-relevant information was found to be distributed across multiple brain regions, how regional nodes in this distributed network reorganize their functional interactions throughout learning remains to be elucidated. Here, we performed chronic, large-scale single-unit recording across 10 cortical and subcortical regions using ultra-flexible microelectrode arrays in mice performing a visual decision-making task and tracked mesoscale functional network dynamics throughout learning. Task learning reshaped interregional functional connectivity, leading to the emergence of a subnetwork involving visual and frontal regions during the acquisition of correct No-Go responses. This reorganization was accompanied by a more widespread representation of visual stimulus across regions, and a region's network rank strongly predicted its peak timing of visual information encoding. Together, our findings revealed that mesoscale networks undergo dynamic restructuring during learning, with functional connectivity ranks influencing the propagation of sensory information across the network.

## Introduction

Learning to make appropriate decisions based on external stimuli is fundamental for survival and adaptive behavior. This process relies on the coordinated activity of distributed neural circuits spanning sensory, association, and motor areas. Previous studies have implicated multiple cortical and subcortical regions in visual task learning and decision-making (*Cruz et al., 2023*; *Wang et al., 2023*; *Wang et al., 2020*; *Peters et al., 2022*; *Liu et al., 2020*; *Makino and Komiyama, 2015*; *Broschard et al., 2023*; *Liu et al., 2023*; *Mukherjee et al., 2021*). In the cortex, anterior regions have been found to play important roles. The medial prefrontal cortex (mPFC) was indispensable for both learning

**\*For correspondence:**
renc@ion.ac.cn (CR);
zhaozt@ion.ac.cn (ZZ)

[†]These authors contributed equally to this work

visual discrimination and maintaining enhanced visual acuity after learning (*Wang et al., 2023*). The secondary motor cortex (M2) encoded sensory history information in a flexible visual decision task, and its inactivation impaired adaptive action selection (*Wang et al., 2020*). Visuomotor learning also promoted visually evoked activity in M2 and anterior cingulate cortex (ACC) (*Peters et al., 2022*). Top-down inputs to the primary visual cortex (V1) are also critical. Orbitofrontal cortex (OFC) projections to V1 were required for learning a visual Go/No-Go task (*Liu et al., 2020*), and retrosplenial inputs to V1 were essential for encoding task-related events (*Makino and Komiyama, 2015*). Among subcortical regions, the dorsomedial striatum was necessary for visual category learning (*Broschard et al., 2023*), and innervation from M2 to the dorsal striatum suppressed inappropriate visual decisions (*Liu et al., 2023*). The mediodorsal thalamus (MDTh) regulated prefrontal signal and noise via distinct circuit mechanisms under different scenarios of decision uncertainty (*Mukherjee et al., 2021*). Although the roles of these brain regions have been well studied individually or in pairwise combinations, how these regions dynamically reorganize their functional interactions as a mesoscale network during the learning of decision-making remains unclear. Addressing this question requires an approach that captures large-scale, longitudinal activity patterns across both cortical and subcortical areas.

Recent advances in large-scale neural recordings have enabled monitoring of activity across multiple brain regions, providing new insights into information representation and transformation at a mesoscale level (*Jun et al., 2017*; *Steinmetz et al., 2021*). It has been revealed that the encoding of sensory, choice, and body motion information is not confined to a single or a few brain regions, but widely distributed across brain regions during visual decision tasks (*Steinmetz et al., 2019*; *Musall et al., 2019*; *Laboratory et al., 2024*). Similarly, sensorimotor transformations during decision-making have been found highly distributed across many brain regions following the learning of a visual change detection task (*Khilkevich et al., 2024*). In a Go/No Go tactile discrimination task, multi-fiber photometry revealed functional networks encompassing basal ganglia, thalamus, neocortex, and hippocampus grow and stabilize upon learning, and during learning, most regions shift their peak activity from the time of reward-related action to the reward-predicting stimulus (*Sych et al., 2022*). In a delayed-response paradigm, learning has been associated with emergence of a specific subnetwork involving layer 2/3 neurons in the anterior lateral motor cortex and posterior parietal cortex, accompanied by sparser global functional connectivity across the dorsal cortex (*Chia et al., 2023*). However, most of these studies have focused on neural dynamics in expert animals or have been restricted to superficial cortical layers, largely due to technical constraints of commonly used techniques such as high-density silicon probes and calcium imaging. These approaches typically provide either limited spatial coverage in depth or lack the longitudinal recording capability across learning. As a result, how cortical-subcortical neural spiking dynamics evolve during task acquisition across broad spatial scales remains poorly understood.

To tackle this question, we utilized uFINE-M (ultra-Flexible Implantable Neural Electrodes for Mouse) arrays (*Luan et al., 2017*) to simultaneously record spiking activity across 10 brain regions in mice learning a visual Go/No-Go task over two to three weeks. The chronic implantation capability of uFINE-M arrays enabled us to track the evolvement of a mesoscale functional network including frontal regions (mPFC, OFC, and ACC), motor cortices (M1 and M2), visual cortices (V1, V2M, and V2L), and subcortical regions (striatum and MD thalamus) throughout learning. By analyzing functional connectivity patterns and information encoding dynamics, we found that learning reshaped interregional communication and accelerated the broadcast of stimulus information throughout the network. These findings provide insights into how distributed brain networks adapt during the acquisition of decision-making skills.

## Results
### High-throughput recording in mice performing a visual Go/No-Go task

To investigate the dynamics of the mesoscale functional network during decision-making task learning, we trained head-fixed mice to discriminate between two visual stimuli (vertical vs. horizontal static gratings) using a Go/No-Go paradigm, which has been used to study local neural dynamics during visual associative learning and decision-making (*Liu et al., 2020*). In this task, mice were required to lick a waterspout in response to Go stimuli (Hit trials) within a specified response window to receive a water reward, while withholding licking for the No-Go stimuli (Correct rejection trials, CR trials) to

avoid timeout punishment (*Figure 1A*). Despite substantial daily fluctuations in task performance (*Figure 1—figure supplement 2*), mice generally gained proficiency over time by learning to make more correct rejection decisions at No-Go stimuli (*Figure 1B*).

To capture neural spiking activity across multiple brain regions throughout task learning, we chronically implanted eight 128-channel uFINE-M arrays into the left hemisphere of each mouse brain and simultaneously recorded from 10 brain regions, including frontal regions (mPFC, OFC, and ACC), motor cortices (M1 and M2), visual cortices (V1, V2M, and V2L), and subcortical regions (striatum and MD thalamus) (*Figure 1C and D*). These regions have been implicated in visuomotor tasks (*Wang et al., 2020*; *Liu et al., 2020*; *Broschard et al., 2023*; *Liu et al., 2023*; *Alsiö et al., 2021*; *Zhang et al., 2014*; *Parnaudeau et al., 2018*) and exhibit dense structural connectivity (*Oh et al., 2014*). The ultra-flexible arrays were guided to place by tungsten wires. We verified the accuracy of implantation depth of this implantation approach in a separate test group of mice (*Figure 1—figure supplement 1*), though we observed a tendency for the arrays to end deeper than expected for cortex regions (142.1±55.2 μm, n=7 shanks) and shallower for subcortical structures (–122.6±71.7 μm, n=7 shanks), the extent of depth error was slight. On average, 532.1±92.5 single units (mean ± SD, n=39 sessions from five mice) were recorded across 1024 channels in each session, with no fewer than 15 units recorded from each region of interest (*Figure 1E and F*). Given the fluctuations in behavioral performance (*Figure 1—figure supplement 2*), we categorized trials in early sessions with low behavioral discriminability (*Wickens, 2001*) (d-prime <2) as 'early stage' data, and trials in late sessions with high behavioral discriminability (d-prime >3) as 'expert stage' data (*Figure 1—figure supplement 2*). Only these data were used in the following analyses.

The average firing rate patterns showed substantial changes during learning (*Figure 2A*), with an overall decrease in firing rate during CR trials and an increase in firing rate during Hit trials. Observing the low number of CR trials in early sessions resulted in noisier traces of firing rate, we accounted for this by reconstructing bootstrap-resampled datasets with only 5 trials for each session in both the early stage and the expert stage. The mean trace of reconstructed datasets again showed overall decrease in CR trial firing rate during task learning (*Figure 2—figure supplement 1*). Moreover, many brain regions exhibited significant changes in their temporal profile of activity in CR trials as learning progressed (*Figure 2A*, *Figure 2—figure supplement 1*).

Based on these observations, we calculated the activity onset timing of each single unit relative to the visual stimulus onset ('Materials and methods') and found that the mean activity onset timing formed a clearer sequential activation pattern across brain regions in expert CR trials (*Figure 2B and C*). This sequential pattern was accompanied by a more temporally compressed activation profile with learning, as evidenced by a significant reduction in the spread of peak activation times across regions (*Figure 2—figure supplement 2* and 'Materials and methods', early: 170.0 ± 160.0 ms vs. expert: 56.67 ± 38.58 ms, mean ± SD, $p < 1.0 \times 10^{-4}$, t-test with Sidak correction). Additionally, several visual and frontal regions (V1, V2M, mPFC, OFC, M2, and M1) also showed faster average response time following stimulus onset in expert CR trials compared to the early stage (*Figure 2D*). This learning-induced compression of mesoscale activity is reminiscent of similar phenomena observed in motor skill learning (*Makino et al., 2017*), suggesting a potential general principle of temporal refinement in distributed brain networks during task acquisition.

In contrast, learning this visual Go/No-Go task did not induce significant changes in the average activity onset timing for most regions in Hit trials (except V2L, *Figure 2D*), nor did it significantly alter the spread of regional peak activation times (*Figure 2—figure supplement 2*). This could be attributed to the fact that the Go stimulus was already introduced during pre-training stages, when the mice learned the basic trial structure ('Materials and methods').

In summary, we found that learning this visual Go/No-Go task led to a sequential activation pattern of the neural activity across brain regions, particularly generating an earlier and more compressed activity sequence in CR trials. These results led us to further explore the leading-following relationships between regions and the roles of individual brain regions within the functional brain network. We focused specifically on CR trials, as mice improved their performance mainly by learning to correctly reject No-Go stimuli.

## Ranking dynamics of mesoscale brain network during learning of CR trials

To study functional connectivity dynamics and quantify the overall extent of leading-following relationship among spiking activity across brain regions, we first identified neuron pairs that exhibited

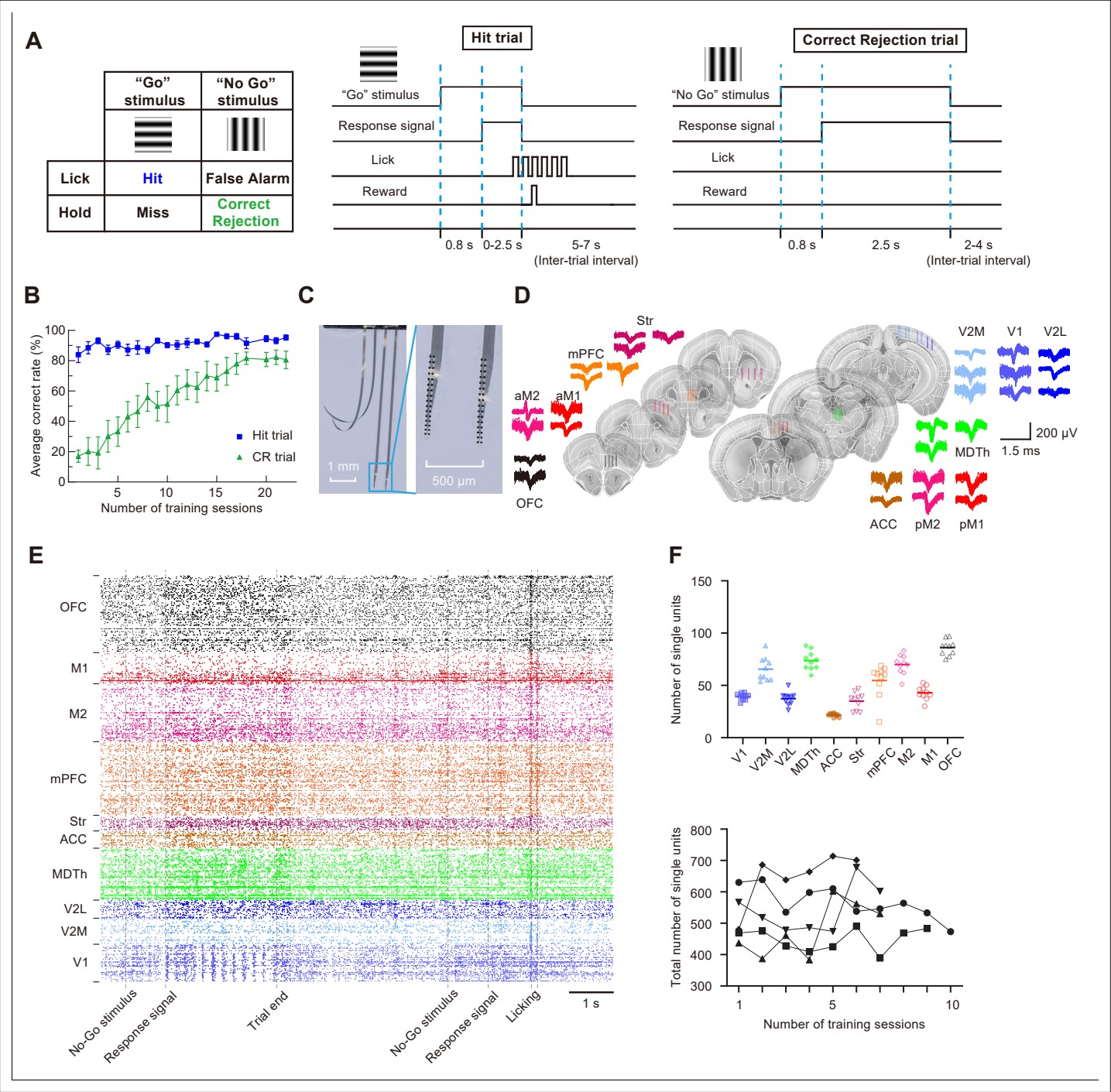

**Figure 1.** High-throughput recording in mice performing a visual Go/No-Go task. (**A**) Schematic of the task. (**B**) Average correct rate during training (mean ± SEM, n = 7 mice). (**C**) Photos showing the uFINE-M shanks and recording sites. (**D**) Schematic showing the implantation sites of uFINE-M arrays, along with example single-unit waveforms recorded from each brain region. Brain section images are taken from The Scalable Brain Atlas (**Bakker et al., 2015**) derived from data in **Lein et al., 2007**. (**E**) Example spike rasters during two trials. (**F**) Top, the number of single units recorded in each brain region. Each data point represents data from an individual recording session. Bottom, the total number of single units recorded during training (n = 5 mice). Each symbol represents data from an individual mouse.

The online version of this article includes the following figure supplement(s) for figure 1:

**Figure supplement 1.** Implantation depth accuracy test.

**Figure supplement 2.** Behavioral performance in daily sessions.

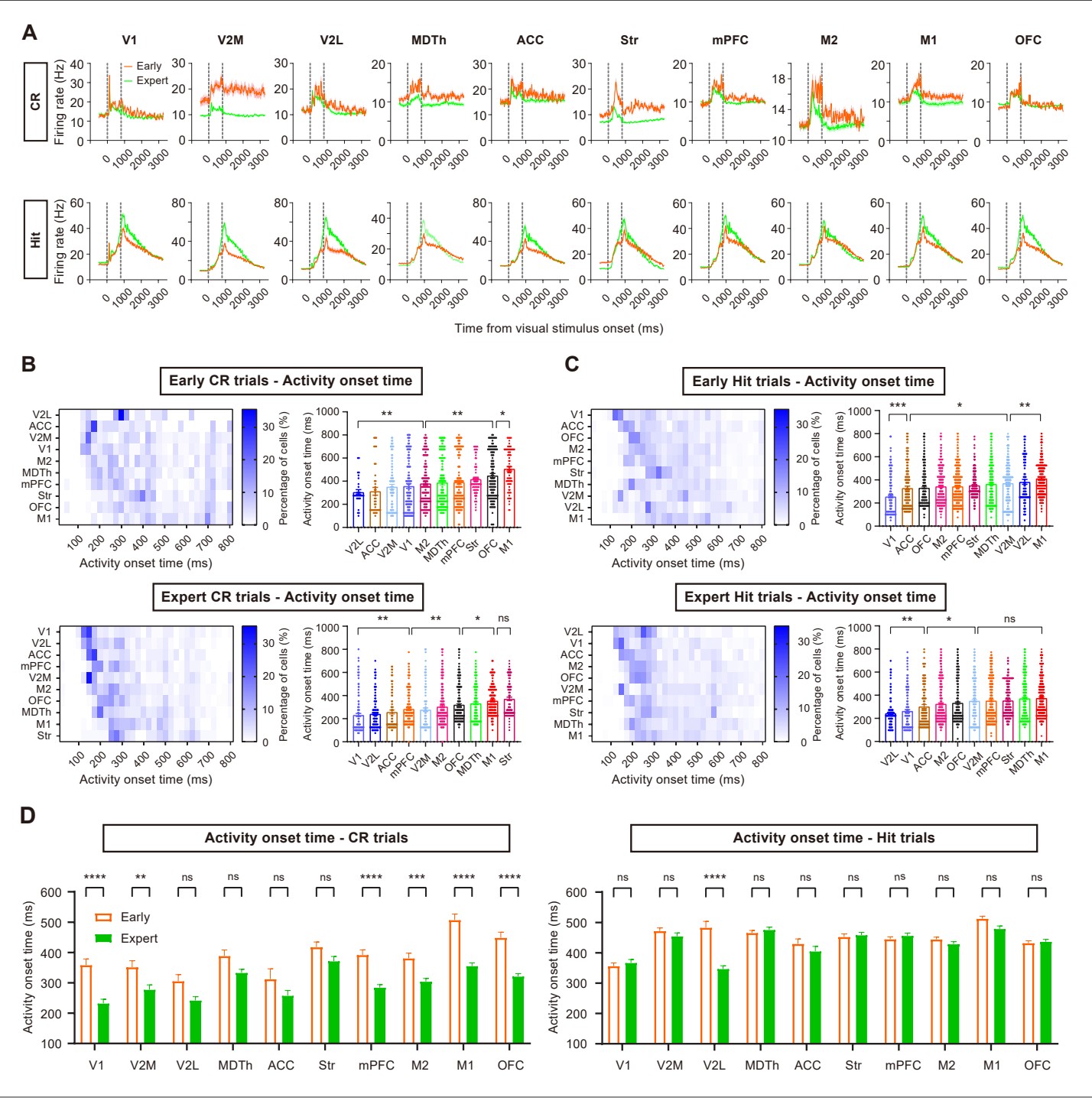

**Figure 2.** Activity changes throughout task learning. (**A**) Averaged firing rate aligned to the visual stimulus onset for all CR trials and Hit trials in the early and expert stages (n = 118 early CR and 828 early Hit trials from 7 sessions of 3 mice, 610 expert CR trials and 677 expert Hit trials from the same mice). Shading, SEM. (**B**) Left, distribution of activity onset timing across time. Right, activity onset timing of each region. Each data point represents data from a neuron. (**C**) Same as B but for Hit trials. (**D**) Comparison of activity onset timing between the early and expert stages. *p < 0.05, **p < 0.01, ***p < 0.001, ****p < 0.0001, t-test with Sidak correction. Error bars, SEM.

The online version of this article includes the following figure supplement(s) for figure 2:

**Figure supplement 1.** Average firing rate from the bootstrap-resampled datasets.

**Figure supplement 2.** Spread of peak activation times across brain regions.

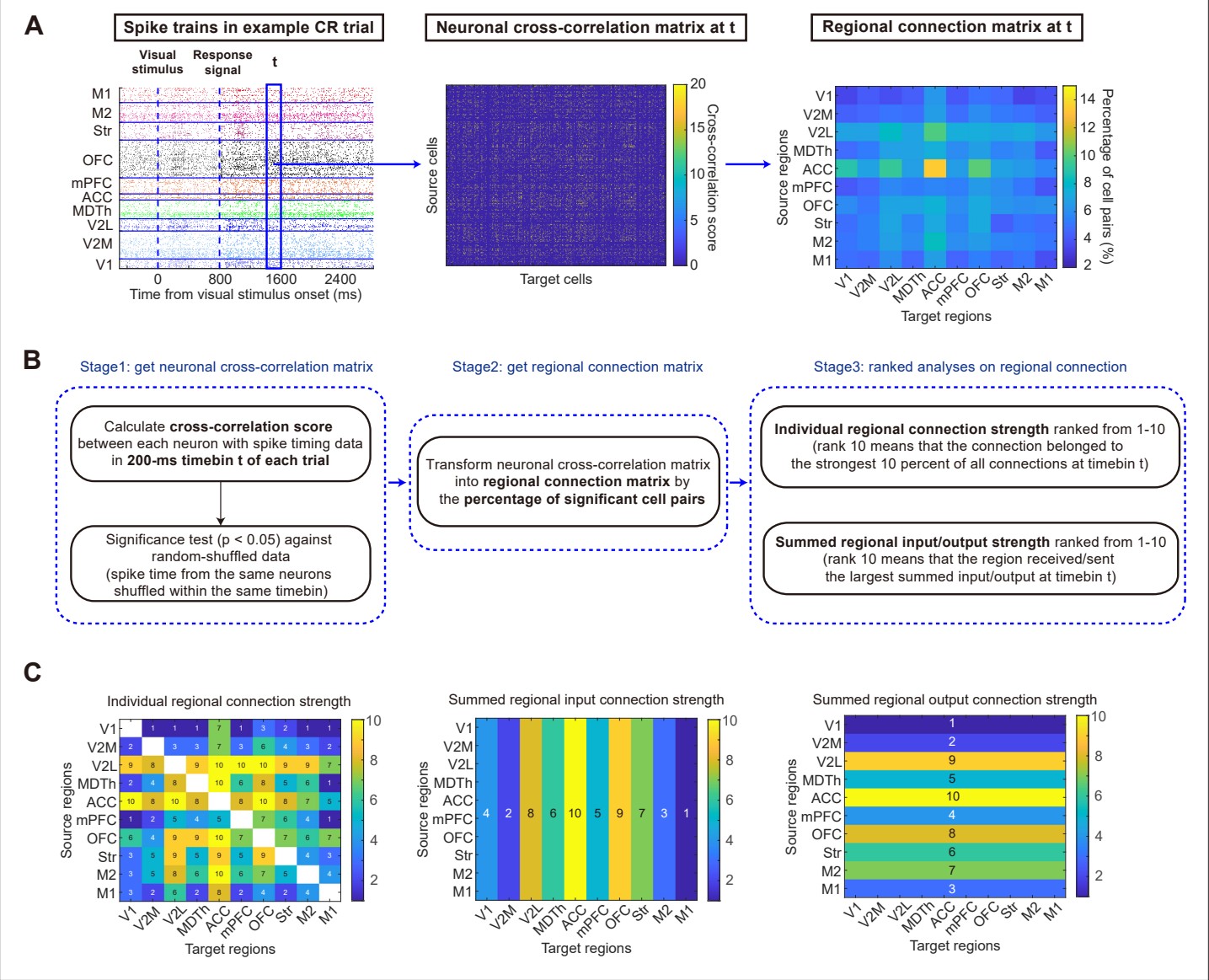

**Figure 3.** Definition of functional connection. (**A**) Schematic of data processing flow of calculating functional connectivity. For each 200-ms time window (t), cross-correlation scores were calculated between spike trains of neuron pairs and the percentage of neuron pairs that showed significant cross-correlations was treated as functional connection strength between brain regions. The regional connection matrix was then ranked from 1 to 10 to evaluate the relative importance of regional connection compared with other connections within the same time window of the same trial. (**B**) Details of processing stages. (**C**) Connection rank matrix calculated from the example data in **A**. The individual regional connection strength measurement (left) was used in *Figure 6*, *Figure 6—figure supplement 1*. The summed regional connection strength measurement (middle and right) was used in *Figures 3–5 and 7*, *Figure 4—figure supplement 1*, *Figure 5—figure supplements 1–3*.

functional connectivity if their cross-correlation score of spiking activity (TSPE algorithm) (*De Blasi et al., 2019*) was above chance level (p<0.05, *Figure 3*). We focused on fast connections within 20ms and included only excitatory connections in subsequent analyses. If the spiking activity of neuron *a* preceded that of neuron *b*, we defined neuron *a* as having functional output to neuron *b*, and neuron *b* receiving functional input from neuron *a*. The functional input/output strength between any two brain regions was then defined as the proportion of neuron pairs with significant excitatory functional input/output, relative to the total number of possible input/output neuron pairs between these two regions. Considering that differences in firing rates might bias cross-correlation between spike trains (*de la Rocha et al., 2007*), making raw counts of significant neuron pairs difficult to compare across conditions, we ranked the values in the regional connection matrix on a scale from 1 to 10.

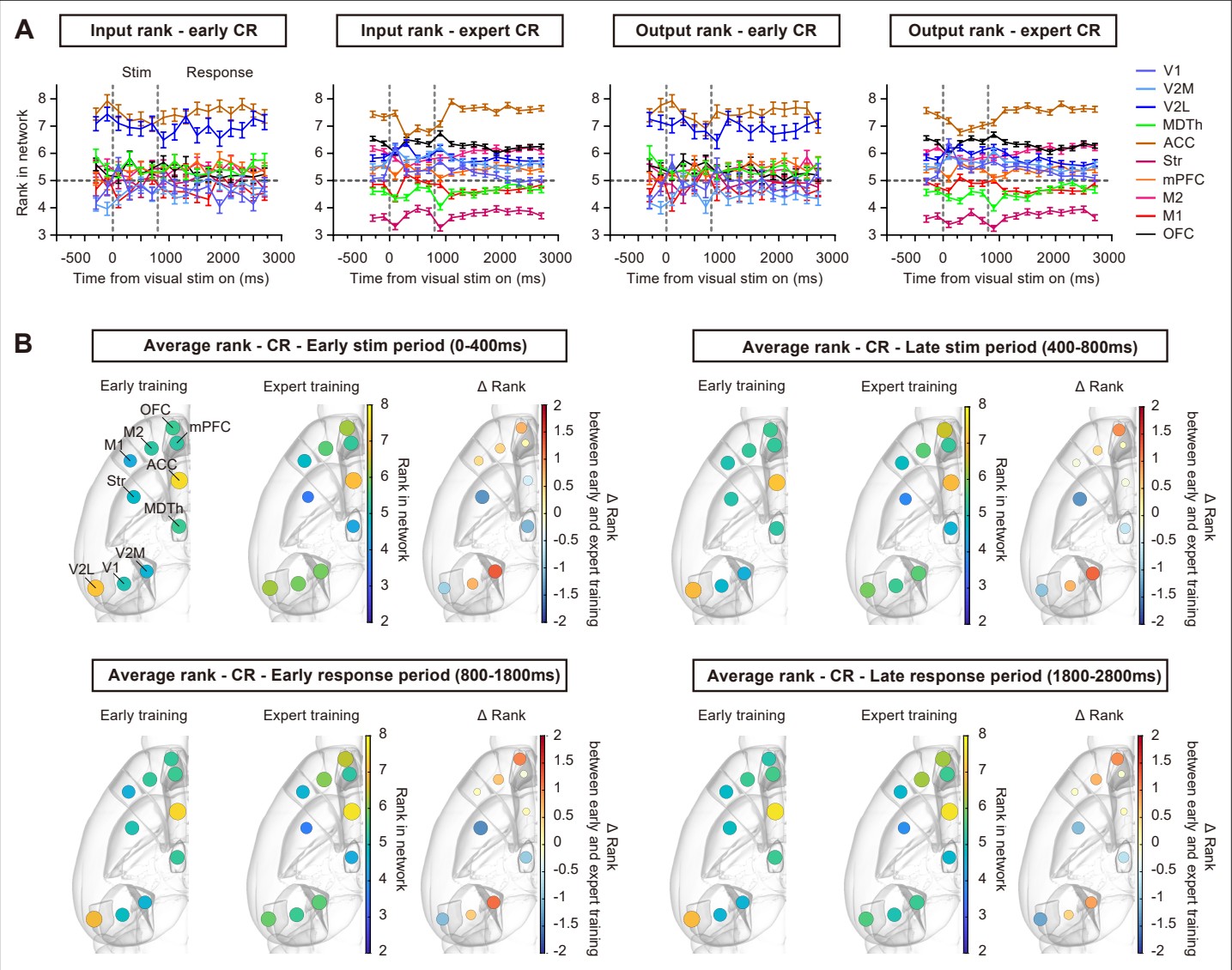

**Figure 4.** Ranking dynamics in CR trials during learning. (**A**) Input/output ranking dynamics during early and expert CR trials. (**B**) Average rank of each brain region in the early stimulus period (0–400 ms after stimulus onset), late stimulus period (400–800 ms after stimulus onset), early response period (800–1800 ms after stimulus onset), and late response period (1800–2800 ms after stimulus onset) of early and expert CR trials, mapped on brain atlas (*Bakker et al., 2015*). ΔRank represents the rank change between the expert and early stages. n = 118 early CR trials from 7 sessions of 3 mice, and 610 expert CR trials from 6 sessions of same mice. Error bars, SEM.

The online version of this article includes the following figure supplement(s) for figure 4:

**Figure supplement 1.** Ranking dynamics in CR trials during learning.

**Figure supplement 2.** Motion energy in CR trials during learning.

This ranking approach enabled us to focus on the relative importance of each region within the brain network and more effectively evaluate the ranking dynamics across time windows and trial types.

In early CR trials, most brain regions did not show obvious differences in input/output rankings (*Figure 4A*), with rank values remaining close to 5 (the expected level for random data) throughout the CR trial. Only the ACC and V2L maintained high rank values in early CR trials, suggesting their roles in visual attention and high-level visual processing. After the mice achieved proficiency in this task, we observed a clear separation of input/output rankings among different brain regions (*Figure 4A*). ACC maintained a high rank, whereas the ranks of V2L, striatum, and MDTh decreased across all trial periods. In contrast, the ranks of V1, V2M, and OFC increased across all trial periods, and M2 exhibited an increased rank during the response period (*Figure 4B*, *Figure 4—figure supplement 1*,

p<0.05, *t*-test with Sidak correction). These results suggest that, during the learning of visual-based decision-making, brain regions within the network differentiate in task involvement. Regions associated with visual processing, value processing, and action selection can emerge as key input/output hubs, forming a more task-relevant subnetwork.

Since previous studies have reported the dominance of movement-related activity across brain regions (*Musall et al., 2019*; *Stringer et al., 2019*), we examined the extent to which the observed changes in functional connectivity patterns could be explained by potential changes in body movements during task training. We performed video recording in a cohort of mice and quantified motion energy of facial movements, foot movements, and pupil dynamics (*Ramseyer, 2020*) during CR trials. Motion energy for face and pupil showed a reduction across all CR trial time windows during learning (*Figure 4—figure supplement 2*), which might account for the overall decrease in firing rates in CR trials (*Figure 2A*), and the decrease for functional connection rank of the striatum. Only feet movements showed a brief increase during the stimulus period (*Figure 4—figure supplement 2*). Thus, the changes in body movements could not fully explain the stable separation of functional connection ranks in the stimulus and responses periods of CR trials during learning.

## Ranking dynamics during learning of Hit trials and fruitless learning

Compared to CR trials, the mesoscale network showed more rapid and dynamic transitions at different intra-trial time points in Hit trials (*Figure 5A*). In early Hit trials, a transient separation of input/output rankings was observed around the visual stimulus onset, with the ACC ranking the highest, which is similar to CR trials. During the response period, regional rankings become more convergent. After the mice reached expert level in this task, although the correct rate of Hit trials remained unchanged (*Figure 1B*), the striatum acquired a high input ranking, particularly during the early response period of expert Hit trials (*Figure 5A and B*, *Figure 5—figure supplement 1A*). The rise of input rank of striatum during the response period was still clear with the analyses repeated on spike time data aligned to the first lick in each Hit trial, indicating this observation was not a result of possible changes in lick initiation time during learning (*Figure 5—figure supplement 2*). Still, we cannot fully rule out the effects from more subtle movement changes during learning, since the motion energy for face also increased in early response period (*Figure 4—figure supplement 2*). In addition to the striatum, we also observed significant changes in the input/output ranks of multiple regions between early and expert Hit trials (*Figure 5—figure supplement 1*), suggesting that distinct mesoscale functional connectivity patterns could underlie similar behaviors.

We also trained mice on a 'fruitless learning' task, in which all visual stimuli and task structure were identical to those in the normal learning group, but the Go/No-Go visual stimuli were presented randomly and had no association with reward. As expected, mice continued to lick in every trial regardless of the visual stimulus type, and we defined the trials in which mice were randomly rewarded as fruitless-learning Hit trials. During the early stimulus period, the ACC and visual regions showed the highest ranks in these fruitless-learning Hit trials (*Figure 5—figure supplement 3*), similar to those in the normal learning group. In the early response period, however, we observed a strong elevation of the rank of the MDTh (*Figure 5—figure supplement 3*), suggesting cognitive effort in the face of uncertainty regarding the task rules (*Marton et al., 2018*; *Lam et al., 2025*). These results further demonstrate that distinct mesoscale functional connectivity patterns can emerge and evolve depending on task demands, in accordance with previous reports (*Chia et al., 2023*; *Cole et al., 2013*; *Pinto et al., 2019*; *Arlt et al., 2022*).

## Rank increase of the visual/frontal regions was attributed to elevated regional connection rank in CR trials

To investigate the factors driving the rank changes in CR trials during learning, we examined the regional connection ranks of brain regions that showed rank increases in CR trials (V1, V2M, M2, and OFC, *Figure 4B and C*, *Figure 4—figure supplement 1*). For each time window within a trial, regional connection strength was ranked on a scale from 1 to 10, with a rank of 1 representing the lowest 10% strength among all regional connections within the same time window (*Figure 3B and C*). We observed a general increase in the ranks of regional connections between these regions (*Figure 6*, V1, V2M, M2, and OFC) and other frontal and motor regions (*Figure 6*, mPFC, ACC, M2, and M1).

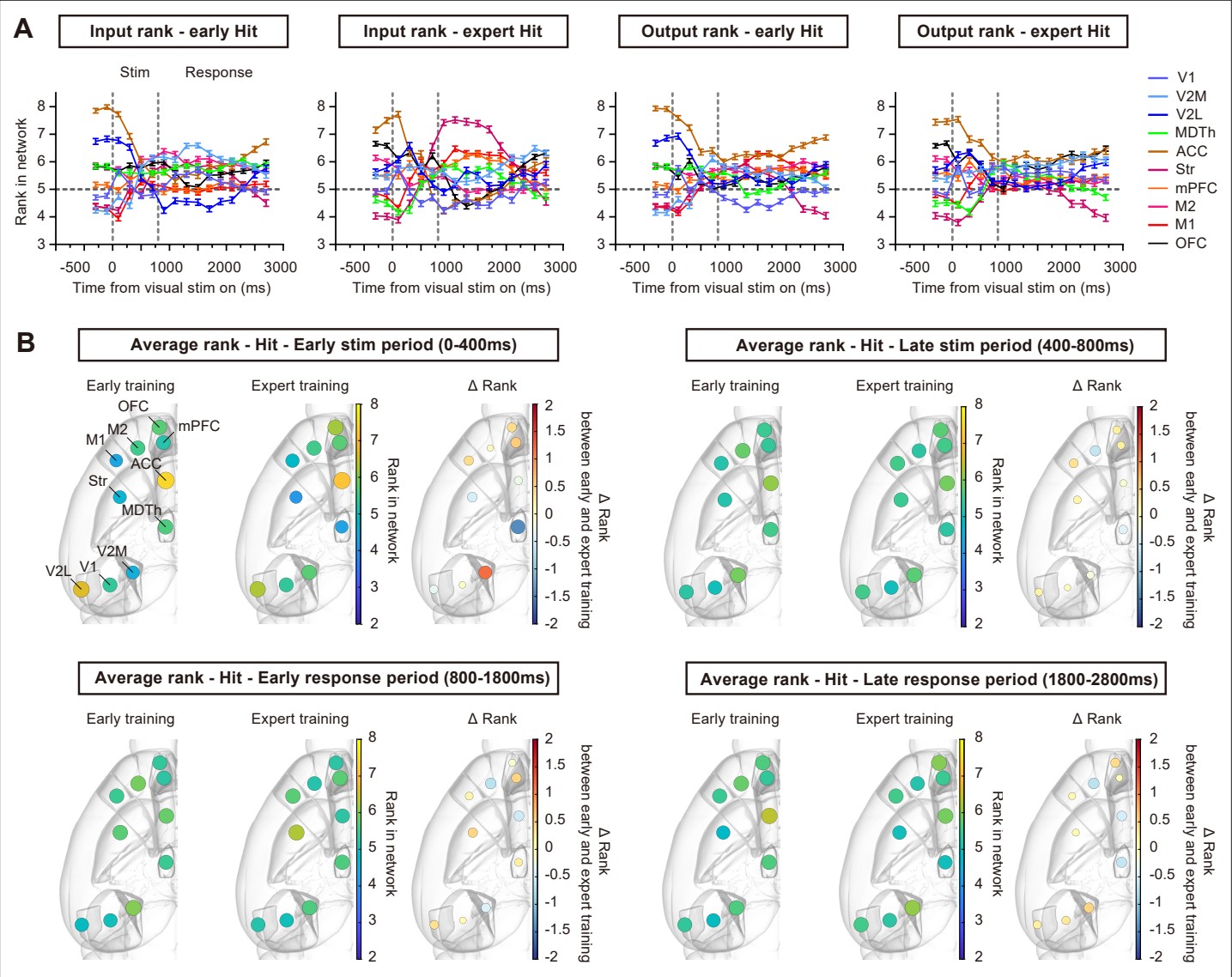

**Figure 5.** Ranking dynamics in Hit trials during learning. (**A**) Input/output ranking dynamics in early and expert Hit trials. (**B**) Average rank of each brain region in the early stimulus period (0–400 ms after stimulus onset), late stimulus period (400–800 ms after stimulus onset), early response period (800–1800 ms after stimulus onset), and late response period (1800–2800 ms after stimulus onset) of early and expert Hit trials, mapped on brain atlas (*Bakker et al., 2015*). ΔRank represents the rank change between the expert and early stages. n = 828 early Hit trials from 7 sessions of 3 mice, and 677 expert Hit trials from 6 sessions of 3 mice. Error bars, SEM.

The online version of this article includes the following figure supplement(s) for figure 5:

**Figure supplement 1.** Ranking dynamics in Hit trials during learning.

**Figure supplement 2.** Ranking dynamics in Hit trials during learning, with spike time aligned to the first lick of each trial.

**Figure supplement 3.** Ranking dynamics in fruitless-learning Hit trials compared to expert Hit trials.

We also examined the regional connection ranks of regions that exhibited rank decreases in CR trials (V2L, MDTh, and striatum, *Figure 6—figure supplement 1*). All three regions showed a decline in regional connection rank both with each other and with most frontal and motor regions (mPFC, ACC, M2, and M1). The striatum, which exhibited the most pronounced rank decrease, showed the most widespread reduction in regional connection ranks.

In summary, these results suggest that the network forms a more compact functional subnetwork during learning to reject the No-Go visual stimulus. This reorganization is characterized by increased relative connection strength among several key visual (V1 and V2M), frontal (mPFC, OFC, and ACC),

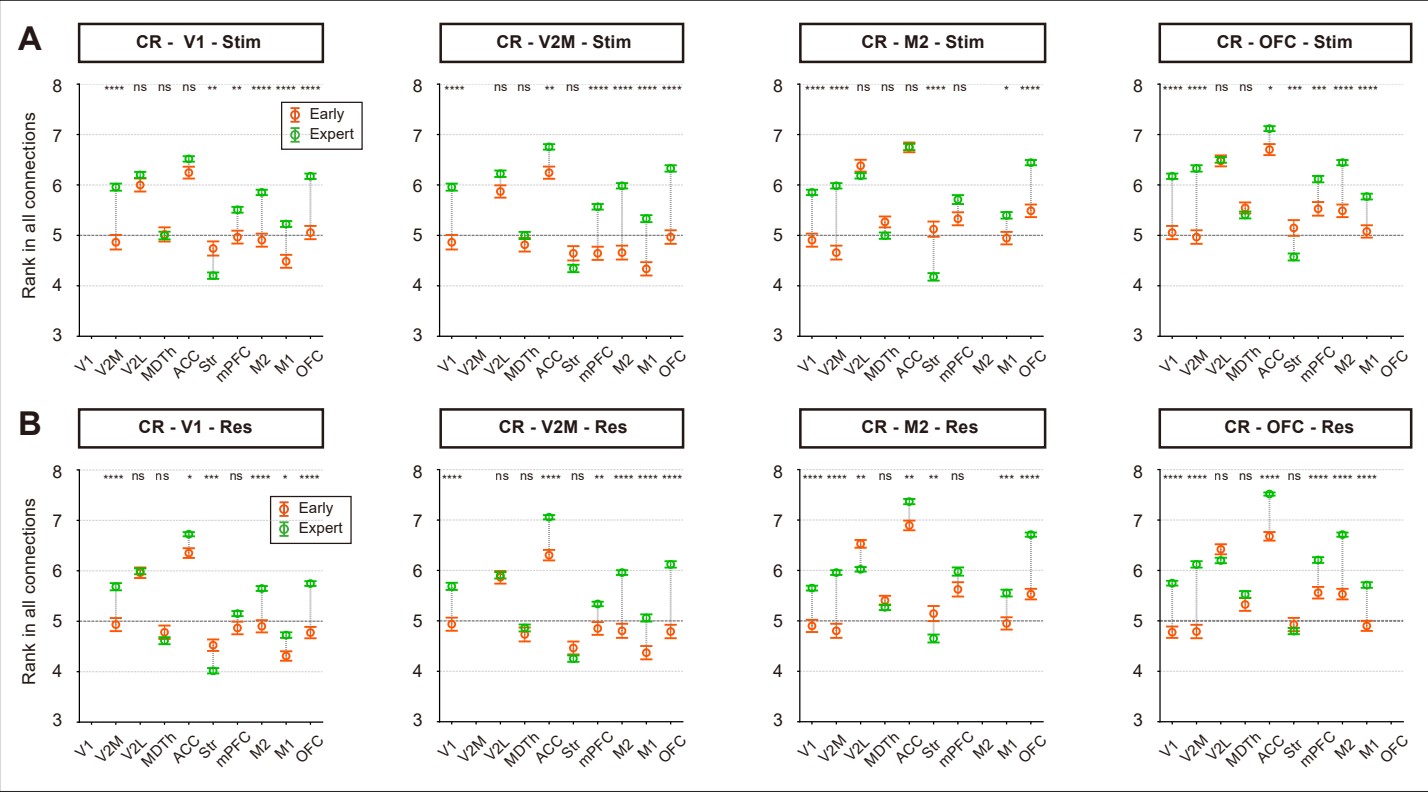

**Figure 6.** Rank increase in CR trials was attributed to elevated input/output rank from/to other regions. (**A**) Average functional connection rank changes for the four regions (V1, V2M, M2, and OFC) that showed increased rank values in the stimulus period of CR trials during visual learning. *p < 0.05, **p < 0.01, ***p < 0.001, ****p < 0.0001, t-test with Sidak correction. (**B**) Same as A but for the response period of CR trials. n = 118 early CR trials from 7 sessions of 3 mice, and 610 expert CR trials from 6 sessions of 3 mice. Dashed lines at rank 5 indicate the average level of random data. Error bars, SEM. Stim, stimulus period. Res, response period.

The online version of this article includes the following figure supplement(s) for figure 6:

**Figure supplement 1.** Rank decrease in CR trials was attributed to reduced input/output rank from/to other regions.

and motor regions (M2 and M1), while regions such as V2L, MDTh, and the striatum become less engaged in the task-related functional network.

## Visual stimulus information became widespread in the stimulus period as learning progressed

After examining network dynamics during different trial periods and learning stages, we wondered how the stimulus encoding ability of each region changed during task learning. To assess the stimulus encoding ability based on spike counts, we grouped trials according to visual stimulus identity (with behavioral choice balanced) and applied receiver operating characteristic (ROC) analyses in each 200 ms time window (*Wickens, 2001*). In each session, spike count data for each neuron was bootstrap-resampled to balance the number of trials across different trial types (*Figure 7A*). For each neuron, 50 trials were resampled with replacement for each trial type to perform ROC analyses, and this procedure was repeated 500 times for each time window. A neuron was classified as stimulus-selective if its ROC selectivity was above 95% of its own randomly shuffled spike count data (p<0.05) in more than 95% of resampling iterations.

In the early training stage, stimulus-selective neurons were mainly found in V1 during the stimulus period (*Figure 7B and C*), while other regions contained very few stimulus-selective neurons during this period. By the late response period, stimulus-selective neurons were found in larger proportions in nearly all regions (except V2L), suggesting that the visual information was broadcast through the network at this time (*Figure 7B and D*). In the expert stage, however, stimulus-selective neurons emerged in all regions during the stimulus period, and their proportion also substantially increased

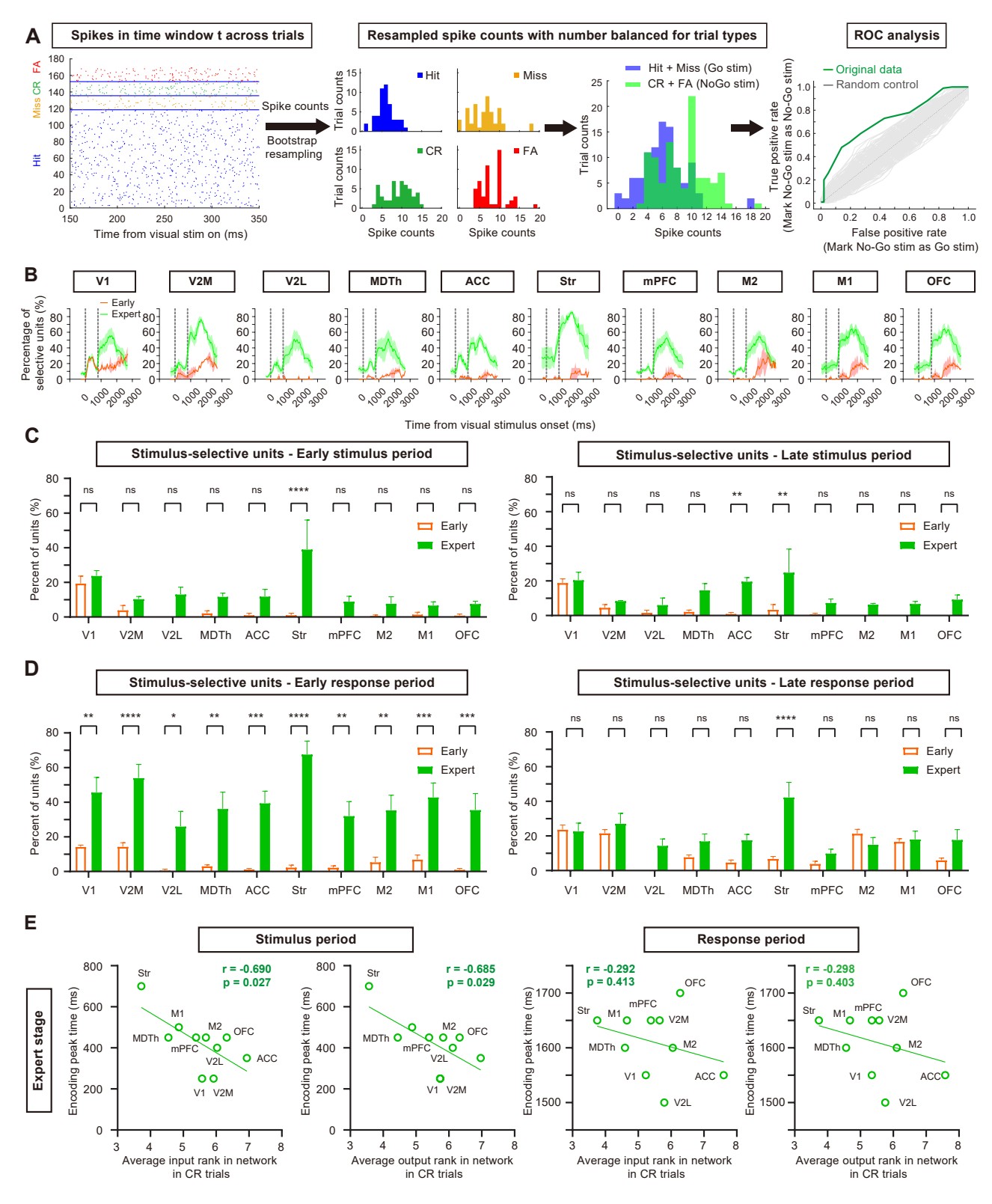

**Figure 7.** Encoding of visual stimulus information during task learning. (**A**) Schematic of the ROC analyses and example data from a neuron preferring the No-Go stimulus. (**B**) Percentage of stimulus-selective neurons in each brain region during the early and late training stages. (**C**) Mean percentage of stimulus-selective neurons in the early (0–400 ms after stimulus onset) and late (400–800 ms after stimulus onset) stimulus periods. n = 10 time bins for the early training stage and 30 time bins for the expert training stage. (**D**) Same as C, but for the early (800–1800 ms after stimulus onset), and late

*Figure 7 continued on next page*

*Figure 7 continued*

(1800–2800 ms after stimulus onset) response periods. *p < 0.05, **p < 0.01, ***p < 0.001, ****p < 0.0001, t-test with Sidak correction. n = 34 time bins for the early training stage and 102 time bins for late training stage. (**E**) Correlation between stimulus encoding peak time and input/output rank in expert CR trials. A significant correlation was observed during the stimulus period but not the response period (Pearson's correlation).

during the response period (*Figure 7D*), in alignment with previous reports that visuomotor learning could promote visually evoked activity in dorsal medial prefrontal cortex (*Peters et al., 2022*), though this could also be the result of the potential movement difference in FA trials and Hit trials (*Figure 4— figure supplement 2*). We also noticed that some regions showed stimulus encoding even before the visual stimulus onset, suggesting effects from trial history (*Marmor et al., 2023*) or expert mice had likely learned the pseudo-random trial sequence ('Materials and methods') and anticipated upcoming visual stimuli based on sensory history.

In expert mice, ROC encoding curves of most regions showed two distinct peaks—one during the stimulus period and another during the response period (*Figure 7B*, except the striatum curve, which ramped up in stimulus period and only showed one peak in response period). Therefore, we defined the peak time of stimulus encoding in each trial period as the center of the time window with the highest mean percentage of stimulus-selective neurons. We found a significant correlation between the input/output rank of each brain region in CR expert trials and its encoding peak time during the stimulus period (p<0.05 Pearson's correlation, *Figure 7E*), with higher-ranked regions reaching their encoding peaks earlier. No significant correlation was observed during the response period in expert CR trials (*Figure 7E*).

In summary, as learning progressed, visual information propagated more rapidly through the network, likely due to the increased functional connection ranks between visual and frontal regions. Moreover, a region's connection rank within the network became highly predictive of how quickly it reached its encoding peak during stimulus viewing.

## Optogenetic inhibition of rank-increasing regions impaired task learning

Finally, to examine whether the regions with increased rank during CR trials actually contributed to task learning, we performed manipulation experiments on two of these regions, specifically V2M and OFC. For each manipulation group, we expressed AAV2/9-mCaMKIIa-eJaws3.0-mRuby3-WPRE-pA (AAV2/9-mCaMKIIa-mCherry-WPRE-pA for the control group) in the bilateral OFC or V2M and inhibited these regions during either the stimulus or response period of task training (*Figure 8*).

We found bilateral inhibition of the OFC (*Figure 8B and C*) showed significantly impaired task learning in both the stimulus period (n=8 mice for eJaws 3.0 group, n=14 mice for mCherry group, $F(1, 399)=29.91$, $p<1.0 \times 10^{-4}$, $\eta^2=0.047$, two-way ANOVA), and the response period (n=8 mice for eJaws 3.0 group, n=16 mice for mCherry group, $F(1, 420)=87.51$, $p<1.0 \times 10^{-4}$, $\eta^2=0.098$, two-way ANOVA). The interaction with training sessions was not significant for both periods ($F(19, 399)=0.75$, $p=0.77$, $\eta^2=0.022$ for the stimulus period, $F(19, 420)=1.05$, $p=0.40$, $\eta^2=0.022$ for the response period), suggesting consistent impairment across training sessions. Bilateral inhibition of V2M during the stimulus period also impaired task learning with a small effect size (n=8 mice for eJaws 3.0 group, n=14 mice for mCherry group, $F(1, 399)=8.19$, $p=4.4 \times 10^{-3}$, $\eta^2=0.012$, two-way ANOVA, *Figure 8F*), whereas inhibition during the response period did not affect task learning (n=8 mice for eJaws 3.0 group, n=16 mice for mCherry group, $F(1, 440)=0.0095$, $p=0.92$, two-way ANOVA, *Figure 8G*). None of the manipulation groups showed significant differences in miss rate compared to the mCherry control group (p>0.05, Welch's *t*-test, *Figure 8D and H*), indicating the observed performance decline was not due to task abandonment.

Taken together, the manipulation effects on task performance provide some support for the connection rank analysis, suggesting that regions with increased rank during learning likely contribute to task acquisition. However, while a rise in connection rank may reflect a region's involvement in the learning process, it does not necessarily imply a causal relationship with learning.

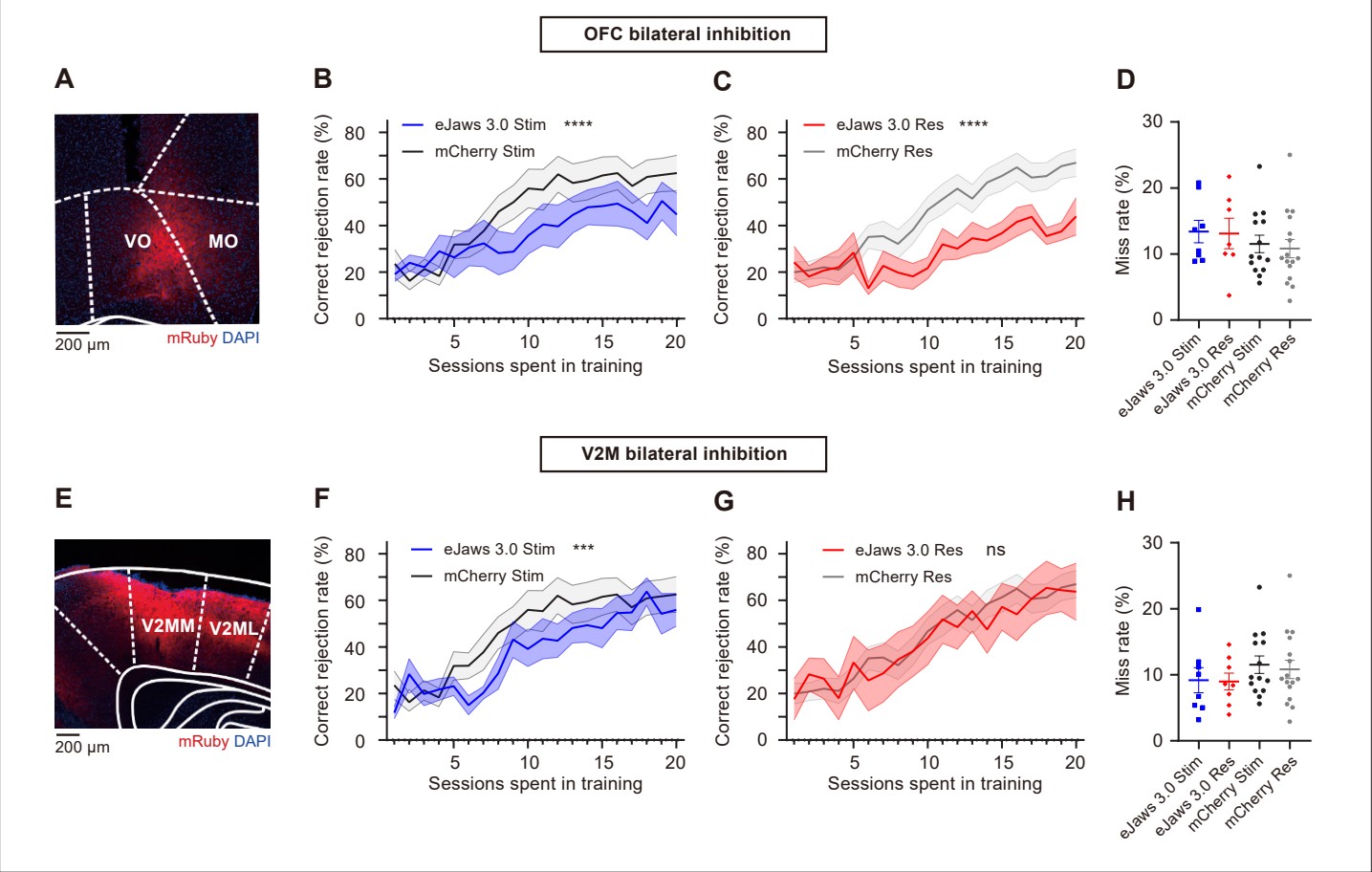

**Figure 8.** The effects of bilateral optogenetic inhibition on task performance. (**A**) Expression of AAV2/9-mCaMKIIa-eJaws3.0-mRuby3-WPRE-pA in the OFC. VO: ventral orbitofrontal cortex; MO: medial orbitofrontal cortex. Regions were named according to the Paxinos atlas (*Paxinos, 2019*). (**B**) Correct rejection rate for the OFC-stimulus period inhibition group (eJaws 3.0 Stim), and the control group (mCherry Stim). Shading, SEM. n = 8 and 14 mice for the eJaws 3.0 and mCherry group, respectively. ****p < 0.0001, significance for the group factor in two-way ANOVA. (**C**) Same as B, but for the OFC-response period inhibition group (n = 8 mice) and control group (n = 16 mice). (**D**) Average miss rate for each mouse in the OFC manipulation group and control group. (**E-H**): Same as A to D but for V2M inhibition. n = 8 mice for each manipulation group. ***p < 0.001, significance for the group factor in two-way ANOVA. The control group here was the same group of mice in **A-D**.

## Discussion

In this study, we investigated how mesoscale functional networks changed during the learning of a visual-based decision-making task. Using 1024-channel uFINE-M arrays for chronic spiking activity recording across multiple cortical and subcortical regions, we were able to examine the mesoscale network dynamics at different timescales: rapid transitions between different periods within a trial, distinct functional connectivity patterns across trial types within a session, and the long-term evolvement of network dynamics throughout task learning.

A key finding of our study is that task learning reshaped interregional connectivity, leading to the emergence of a more task-relevant subnetwork as mice learned to correctly reject No-Go stimuli. Specifically, several visual and frontal regions (V1, V2M, OFC, and M2) gained prominence in the network, while others (V2L, MDTh, and striatum) became less engaged. These findings suggest that learning is accompanied by a selective refinement of interregional communication, with a shift in functional connectivity toward regions more directly involved in processing task-relevant information, and shifts in peak activity time to form a faster and more compressed activity sequence across regions. These observations align with previous reports of the dynamic reorganization of cortico-basal ganglia-thalamo-cortico network during the learning of a tactile discrimination task (*Sych et al., 2022*), the spatiotemporal refinement in cortical activity during the learning of a texture discrimination task (*Gilad and Helmchen, 2020*) and a visually guided delayed-response task (*Chia et al., 2023*), and the

enhanced coupling between somatosensory neurons and frontal neurons in a whisker detection task (*Esmaeili et al., 2022*), suggesting that the emergence of a more task-relevant functional network may be a general mesoscale network feature of learning.

Beyond connectivity changes, we also found that the encoding of stimulus information became more widely distributed across the network as learning progressed. Moreover, the connectivity rank of a brain region was strongly correlated with the timing of its stimulus encoding peak during the stimulus period, suggesting that high-ranked regions may not only receive information earlier but also play a more central role in relaying task-relevant signals. These findings indicate that learning facilitates more efficient information flow through the network, potentially enhancing sensory processing and decision-making processes. The broader recruitment of stimulus-selective neurons during the response period in expert mice further supports the notion that learned associations between sensory inputs and behavioral outcomes become increasingly embedded in distributed circuits over time (*Laboratory et al., 2024*; *Khilkevich et al., 2024*).

We also noticed discrepancies between the results of network ranking analyses and optogenetic inhibition experiments. Inhibiting OFC during either the stimulus or response period significantly impaired learning, consistent with its increased rank in both task periods. However, while V2M also showed increased network rank in both the stimulus and response periods, inhibition of V2M during the response period had no significant effect. This suggests that rank increases with learning do not necessarily indicate a direct causal role in driving behavioral improvements. Other factors, such as neuromodulatory influences and internal state changes, may also contribute to the observed changes in functional connectivity (*Shine, 2019*). We also cannot fully rule out the possible effect from rebound activity following optogenetic inhibition (*Li et al., 2019*; *Parrish et al., 2023*), which may confound the interpretation of manipulation effects during the stimulus period (*Figure 8*), but did not change the causal role of OFC in learning and the discrepancies between ranking analysis and the manipulation results in this case.

Despite recording from 10 brain regions, our study remains limited compared to the extensive network of brain regions implicated in visual-based decision-making (*Laboratory et al., 2024*), including the midbrain, hindbrain, and cerebellum. The limited anatomical coverage and relatively simple task design might restrict the generalizability of our findings to more complex forms of decision making, and the observed changes in ranking dynamics could also arise from broader shifts in arousal, attention, or motivation over repeated sessions. We also didn't succeed in chronically tracking enough number of neurons for each region, which denied the chance to investigate single neuron level functional connection changes during learning. Considering that differences in firing rates might bias cross-correlation between spike trains (*de la Rocha et al., 2007*) and making raw counts of significant neuron pairs difficult to compare across conditions, we focused on the relative importance of each region within the brain network and took a ranking approach to more effectively evaluate the ranking dynamics across time windows and trial types. But this approach remained descriptive and might obscure magnitude of differences in connection strengths. Finally, though we balanced the number of each trial type in the encoding analyses, we cannot fully rule out the influence from differences in movements between Hit trials and FA trials, which might explain the large percentage of encoding neurons in the late response period. There has been extensive literature (*Steinmetz et al., 2019*; *Musall et al., 2019*; *Stringer et al., 2019*; *Gilad et al., 2018*) on the strong effects of body movements on brain dynamics, and we also found the motion energy dynamics of the mice could explain the broad decrease in CR trial firing rates, and the decline in functional connection rank of the striatum. Though we found a decrease in motion alone cannot fully explain the development of functional connection dynamics across the network, since the mice used in motion analyses were from a separate group, with only limited body parts monitored during learning, we could not further disentangle movement-related neural activity from task-related signals. Given these limitations, future studies should aim for broader spatial coverage, ideally with stable tracking of the same neuronal populations throughout the entire learning process, to achieve a more comprehensive characterization of the mesoscale network dynamics. Additionally, carefully designed decision-making tasks (*Aguillon-Rodriguez et al., 2021*) will be essential for disentangling neural representations of sensory stimulus information, decision-making, action execution, and arousal states. It is also essential to establish more reliable analysis approaches that could provide more accurate assessment of functional connection dynamics without bias from firing rates. More importantly, as recording scales continue to expand,

future work should aim to systematically evaluate the predictive power of different analysis methods in determining the causal contributions of various brain regions to learning and decision-making.

## Materials and methods

### Animals

Animal use procedures were approved by the Animal Care and Use Committee at the Center for Excellence in Brain Science and Intelligence Technology, Chinese Academy of Sciences (approval number NA-056-2023). Data were collected from a total of 87 male adult C57BL/6 mice (3–5 months old). Among them, 7 were used for task training to acquire behavioral learning results without electrode array or optical fiber implants, 5 for electrophysiological recordings during behavioral task (three for visual-based decision-making learning and 2 for fruitless learning), 69 for optogenetic manipulation experiments, 3 for video analysis, and 3 for verification of implantation approach. Mice were generally housed in groups of 3–4 per cage, but mice for chronic extracellular recordings were housed individually to protect the implants. Mice were water-deprived in the home cage and received water reward during daily behavioral sessions. On days when mice did not perform the task, restricted water access (~0.8 mL per mouse) was provided each day. All mice were maintained on a 12 h light/12 h dark cycle (lights on at 7:00 a.m.), and all sessions were performed in the light phase.

### Visual Go/No-Go task

Mice were head-fixed during training sessions and positioned in an acrylic tube placed in a behavioral chamber. A capacitive lick detector and a peristaltic water pump were controlled by custom MATLAB (MathWorks) scripts and digital I/O devices (Arduino Uno R3, Arduino) to monitor tongue licks and deliver water reward, respectively. Visual stimuli were presented on a 19" LCD monitor (Dell P1917S, max luminance 80 cd/m$^2$) placed 10 cm from the right eye of the head-fixed mouse. A yellow light-emitting diode (LED) was placed above the waterspout to signal the onset of the response period (response signal).

Each trial was initiated automatically after the preceding inter-trial interval (ITI) expired. A full-field visual stimulus (vertical or horizontal static gratings with spatial frequency of 0.09 cycles/° and 100% contrast) was presented on the monitor for 800 ms, followed by illumination of the yellow LED to indicate the start of the response window (response signal). Mice were required to lick during the response period of 'Go' stimuli to receive a water reward (Hit trials). Failure to respond by licking during the response window of 'Go' stimuli would result in a Miss trial with no reward or punishment. Licking during the response period of 'No-Go' stimuli would be punished with an 8 s timeout (False Alarm trials, FA). Correctly withholding licking for 'No-Go' stimuli (Correct response trials, CR) would be rewarded with a 2 s reduction in the ITI. ITIs were randomized between 4–6 s, but licking during the ITI would extend the interval by an additional 4–6 s (up to a maximum of 30 s) to punish impulsive licking. The training session was terminated if there were no lick responses in 20 consecutive trials.

The mice were trained to perform this task in three sequential steps. In step 1 (days 1–2), mice were allowed to collect water rewards by simply licking the waterspout placed under nose, with a fixed interval of 4 s. In step 2 (days 3–5), mice were required to lick specifically during the response window (at the presence of the LED response signal) and refrain from impulsive licking during the ITI. Only the 'Go' stimulus was presented in step 2. In step 3 (days 6–20), the 'No-Go' stimulus was introduced, and mice were trained to withhold licking for 'No-Go' stimulus to avoid timeout punishment (*Figure 1A*). The trial sequence was pseudo-randomized to maintain a balanced number of 'Go' and 'No-Go' stimuli in every six trials.

For mice used in electrophysiological recordings, each mouse underwent 7.67±2.08 (mean ± SD) sessions in training step 3 until they reached an average correct rate of ~85% in daily sessions. For optogenetics manipulation experiments, mice completed a fixed 20-session training in step 3.

### Design, fabrication, and assembly of ultra-flexible microelectrode array (uFINE-M)

Each microelectrode array contained 128 channels for electrophysiological recording (*Figure 1C*). The array consisted of four flexible implantable shanks, each with 32 recording sites arranged in a 16×2 matrix. Each recording site was circular, with a diameter of 20 μm. The flexible shank was 6 mm in

length, and the longitudinal spacing of electrode recording sites was either 30 μm or 50 μm to suit different spatial coverage requirements. The shank spacing was customized by adjusting the spacing between the four shuttling tungsten wires, which were used to guide the flexible shanks into brain tissue, ensuring proper alignment with the targeted implantation region (*Figure 1D*).

The fabrication of uFINE-M was adapted from a planar microfabrication technique featuring a multilayer architecture, as previously described (*Luan et al., 2017*). The structural and passivation material was non-photosensitive polyimide, and patterning was achieved through $O_2$ plasma etching. Titanium was used as the adhesion layer between metal and polymer. The overall device thickness was limited to 1–1.5 μm to maintain low bending stiffness for minimized tissue damage. Both the interconnects and recording site surfaces were made of gold. A 20 μm diameter hole was designed at the tip of the flexible shank, in which the tip of the shuttling tungsten wire was anchored to drag the shank into brain tissue during implantation. The recording sites were coated with either 200 nm of sputtered iridium oxide film or electrochemically deposited PEDOT:PSS (poly(3,4-ethylenedioxythiophene) polystyrene sulfonate) to lower the electrode impedance to below 100 kΩ at 1 kHz in saline solution.

Each array was soldered to a 128-channel flexible printed circuit (FPC) board measuring 42 mm in length, which was connected to the SpikeGadgets 128-channel headstage (SpikeGadgets, San Francisco, USA) for signal acquisition. The four shuttling tungsten wires were fixed onto a carrier chip with 5% Poly (ethylene oxide)–300000 (PEO; CAS No. 25322-68-3) before implantation.

## Surgery

### Electrode array implantation

Electrode array implantation and viral injection for optogenetic inactivation experiments were performed before behavioral training. Mice were anesthetized with isoflurane before surgery (3–4% for induction, ~1% for maintenance) and head-fixed in a stereotaxic apparatus. Body temperature was maintained at 37°C using a heating pad. Chlortetracycline hydrochloride eye ointment was applied to prevent corneal drying. A circular piece of scalp was removed to expose the skull, and the incision site was treated with cyanoacrylate tissue adhesive (Vetbond, 3M, Saint Paul, USA).

For chronic implantation of uFINE-M arrays (*Figure 1D*), three craniotomies (~6 mm² each) were performed over the left hemisphere, and the dura was left intact. A grounding silver wire was implanted posterior to lambda on the right hemisphere. The cortical surface was kept moist with artificial cerebrospinal fluid or 1×phosphate-buffered saline (PBS). Arrays were implanted to OFC (orbitofrontal cortex, LO, VO and MO, AP 2.46 mm, ML 0.65 mm, depth 2.20 mm), across anterior M1 (primary motor cortex) and anterior M2 (secondary motor cortex, AP 1.94 mm, ML 1.50 mm, depth 0.80 mm), mPFC (medial prefrontal cortex, PL and IL, AP 1.78 mm, ML 0.30 mm, depth 2.20 mm), striatum (caudate putamen, AP 1.42 mm, ML 0.30 mm, depth 2.90 mm), across posterior M1, posterior M2, and ACC (anterior cingulate cortex, Cg1 and Cg2, AP –0.20 mm, ML 0.55 mm, depth 1.20 mm), MDTh (Σ mediodorsal thalamus, MDL, MDC, MDM, AP –1.34 mm, ML 0.30 mm, deep 3.30 mm), across of V1 (primary visual cortex) and V2L (secondary visual cortex lateral area, AP –2.80 mm, ML 3.25 mm, deep 0.90 mm), and V2M (secondary visual cortex medial area, V2MM and V2ML, AP –2.80 mm, ML 1.40 mm, deep 0.90 mm). Brain regions were named according to *The Mouse Brain in Stereotaxic Coordinates* by Franklin and Paxinos (3rd edition) (*Paxinos, 2019*). The shuttling tungsten wires were released from the carrier chip by applying saline to the 5% Poly (ethylene oxide)–300000 fixation site, and the tungsten wires were retracted from the brain 2 min after the implantation. The exposed parts of arrays were bonded together layer by layer using light-curable resin (Filtek Z350 XT, 3M, Saint Paul, USA). The craniotomy was sealed with a thin layer of silicone elastomer Kiwi-Cast (World Precision Instruments, Sarasota, USA). A custom-designed headplate was positioned on the skull and secured using Super-Bond C&B (SUN MEDICAL, Japan). After the Super-Bond Polymer cured, several layers of dental acrylic cement were applied to secure the entire implant.

### Viral injection

For viral injections, the skull was not cracked but only thinned to allow smooth entry of a borosilicate glass pipette with a tip diameter of ~40–50 μm. A total of 150 nL viral solution—either AAV2/9-mCaMKIIa-eJaws 3.0-mRuby3-WPRE-pA (for manipulation group mice, TaiTool Bioscience, Shanghai, China) or AAV2/9-mCaMKIIa-mCherry-WPRE-pA (for control group mice, TaiTool Bioscience, Shanghai, China) was injected at a depth of 1750 μm for OFC and 500 μm for V2M using a

syringe pump (Nanoject II Auto-Nanoliter Injector, Drummond Scientific Company, USA). Group identity (manipulation or control) was randomly assigned among cage mates. After injection, the pipette was left in place for 10–15 min before retraction. Optical fibers were implanted bilaterally above the virus injection sites (1000 μm deep for OFC and on cortical surface for V2M), angled ~10° laterally. Mice were given carprofen (5 mg/kg) subcutaneously for postoperative analgesia.

Mice were allowed to recover from the surgery for at least 3 weeks before water restriction and behavioral training.

## Electrophysiological recording

Neural signals were amplified and recorded using a SpikeGadgets 1024-channel system (SpikeGadgets, San Francisco, USA). Raw voltage signals were sampled at 30 kHz. Task-related behavioral events were digitized as TTL signals and recorded simultaneously by the SpikeGadgets system.

## Optogenetic inactivation

Optical silencing via activation of eJaws 3.0 activation was induced by LED red laser (625 nm; Thorlabs) and controlled by digital I/O devices (Arduino Uno R3, Arduino). To manipulate neural activity in either the OFC or V2M, the laser was delivered during the stimulus period (0–800 ms after visual stimulus onset) or response period (800–2500 ms after visual stimulus onset) of all trials in separate manipulation groups. The laser power at the fiber tip was calibrated to 2 mW.

## Histology

Mice were deeply anesthetized with isoflurane followed by an intraperitoneal injection of 15% ethyl carbamate solution. Transcardial perfusion was then performed using 4% paraformaldehyde (PFA). Brains were extracted, post-fixed in 4% PFA at 4°C overnight, and then transferred to 30% sucrose in PBS until equilibration for cryoprotection. Brains were sectioned at a thickness of 25 μm, and slices were mounted with antifade mounting medium (with DAPI). Fluorescence images were acquired using a virtual slide microscope (VS120, Olympus, Shinjuku, Japan; *Figure 8* and *Figure 1—figure supplement 1*).

## Analysis of behavioral performance

To classify behavioral trial by task performance, we used the d-prime (d') metric (*Wickens, 2001*) and labeled each trial by the d-prime value (*Figure 1—figure supplement 2*) calculated with the 10 trials before and 10 trials after it:

$$d^{'} = norminv(Hit\ rate) - norminv(FA\ rate),$$

where *norminv* is the inverse of cumulative normal function, *Hit rate* is the frequency of Hit response in the Go stimulus trials, and *FA rate* is the frequency of False Alarm response in the No-Go stimulus trials. To minimize confounding effects of animals' motivation on the evaluation of task performance, the trials after the last lick in daily sessions were discarded (*Figure 1—figure supplement 2*).

Mouse oral-facial movements during training were recorded at 250 frames per second (fps) using a high-speed camera (MV-CA016-10UC, Hikrobot Co., Ltd., China). The video data were processed with the open-source software Facemap (*Syeda et al., 2022*) and custom MATLAB scripts. ROIs were manually defined and the motion energy (*Figure 4—figure supplement 2*) at each timepoint was calculated as the absolute value of the difference between consecutive frames, summed across all pixels within ROI (*Stringer et al., 2019*). To account for motion energy changes caused by environment luminance changes (e.g., LED response signal), we subtracted motion energy data of miss trials from the data of other trial types during the response period, as mice did not show observable oral-facial motion in miss trials.

## Analysis of neuronal responses
### Spike sorting
Spike sorting was performed offline using custom MATLAB scripts and open-source software Spyking Circus (*Yger et al., 2018*). Raw voltage signals were first filtered above 300 Hz, then denoised with manual threshold and common-median referenced within each probe shank (*Rolston et al., 2009*).

The preprocessed signals were fed to Spyking Circus, which applies automated density-based clustering and template-matching algorithm for spike detection. Spike detection threshold of each recording site was defined as six times the median absolute deviation of the signal. Spatial whitening was then performed to remove spurious spatial correlation between nearby recording sites, and the spike detection thresholds were recalculated after whitening. The putative spike waveforms were aligned on local minima, projected to five-dimensional feature space with principal component analysis, and the spike templates (putative cells) were constructed with density-based clustering. The temporal width of the spike templates was set as 3 ms, the spatial width was set as 200 μm. The templates were finally matched to the data with an iterative greedy approach, with an assumption that templates sum linearly to find the spike times. The results from Spyking Circus were manually curated using Phy (https://github.com/cortex-lab/phy, *Rossant et al., 2025*) to remove obvious artifacts with abnormal waveform shape and merge similar spike clusters in feature space. Spike clusters were considered as single units if the inter-spike interval exceeded 1 ms and the clusters were well isolated in feature space.

## Firing rate and activity onset timing

To calculate the average firing rate, the spikes were first binned at 1 ms resolution and resolution spike rate was computed over a 25 ms time window. To account for the higher noise in CR trials due to low trial number in some sessions, we constructed bootstrap-resampled datasets for 500 times, with only five trials sampled with replacement for each session in both the early stage and the expert stage to test if the low number of trials affects the results (*Figure 2—figure supplement 1*).

To identify the activity onset timing of individual single units, the firing rate of a single unit in each 25 ms time window across trials was compared to its baseline activity (500–0 ms before the visual stimulus onset) by *t*-test. We identified time windows in which p values were below 0.05 for at least three consecutive time windows and defined the first time window as the timing of activity onset (*Figure 2*).

For each brain region, the time window with the highest proportion of neurons exhibiting activity onset was defined as the regional peak activation time, and the pairwise differences in peak activation times across all brain regions were used as a measure of the temporal compression of activation sequence (*Figure 2—figure supplement 2*).

## Connection rank analyses

Functional connections between neurons were defined based on the significance of cross-correlation scores between their spike trains. For each 200 ms timebin, cross-correlation scores between neurons were calculated with the total spiking probability edge (TSPE) algorithm (*De Blasi et al., 2019*), in which an edge filter was applied to the cross-correlogram to facilitate the detection of local maxima and minima. A functional connection was identified if its cross-correlation score exceeded at least 95% of cross-correlation scores calculated from randomly shuffled spike trains. Only excitatory connections within 20 ms were included in subsequent analyses.

To establish regional connection profiles and identify key brain regions within the network, we defined the functional input/output strength between any two brain regions as the proportion of neuron pairs that had significant excitatory functional connections, of all possible input/output pairs between these two regions. To better evaluate the relative importance of each region within the brain network, we ranked the summed values of input/output strength of each brain region on a scale from 1 to 10 (*Figures 3–5*, *Figure 7*, *Figure 4—figure supplement 1*, *Figure 5—figure supplements 1–3*). To better compare interregional input/output strength, for each time window within a trial, regional connection strength was ranked on a scale of 1 to 10, with a rank of 1 representing the lowest 10% strength among all regional connections within the same time window (*Figure 6*, *Figure 6—figure supplement 1*).

## ROC analysis

To quantify the selectivity of each neuron for visual stimulus, we applied the receiver operating characteristic (ROC) analysis (*Wickens, 2001*) to the distributions of spike counts on each 200-ms time window within the trial. A neuron was included in the ROC analysis only if it had at least five trials for each of the four trial types (Hit, CR, Miss, and FA). For each neuron, 500 bootstrap-resampled data were generated in each time window, and the number of trials with different choices and visual stimuli

were balanced to ensure 50 trials for each condition. The area under the ROC curve (auROC) indicates the accuracy with which an ideal observer can correctly classify whether a given response is recorded in one of the two conditions. ROC selectivity was defined as 2×abs(auROC–0.5), which ranges from 0 to 1. A neuron was classified as stimulus-selective if its ROC selectivity was larger than 95% of randomly shuffled data (p<0.05) in at least 95% of bootstrap-resampled dataset (*Figure 7*).

## Statistical analysis

No statistical methods were used to predetermine sample sizes. Sample sizes were consistent with similar studies in the field. Statistical analyses were performed using MATLAB or GraphPad Prism (GraphPad Software). The two-way ANOVA was used to determine the significance of the effects. Correlation values were computed using Pearson's correlation. Unless otherwise specified, data were reported as mean ± SEM and statistical significance was set at p<0.05.

## Acknowledgements

The authors thank the Nanofabrication Facility for Advanced Brain Science at CEBSIT and Dr. Xiaocheng Li for supporting electrode fabrication and thank Dr. Muming Poo and Dr. Jun Yan for discussion and advice on various details in task design and data analysis. This work was supported by the National Science and Technology Innovation 2030 Major (No. 2021ZD0202200 and No. 2021ZD0202202), Shanghai Municipal Science and Technology Major Project (No. 2021SHZDZX), Lingang Laboratory (No. LG202105-01), the National Natural Science Foundation of China (No. 32200917), and Shanghai Pujiang Program (No. 23PJ1414400).

## Additional information

### Funding

| Funder | Grant reference number | Author |
|---|---|---|
| Ministry of Science and Technology of the People's Republic of China | No. 2021ZD0202200 | Zhengtuo Zhao |
| Ministry of Science and Technology of the People's Republic of China | No. 2021ZD0202202 | Zhengtuo Zhao |
| Shanghai Municipal People's Government | No. 2021SHZDZX | Zhengtuo Zhao |
| Shanghai Municipal People's Government | No. 23PJ1414400 | Chi Ren |
| Shanghai Municipal People's Government | LG202105-01 | Zhengtuo Zhao |
| National Natural Science Foundation of China | No. 32200917 | Zhengtuo Zhao |
| Ministry of Science and Technology of the People's Republic of China | 2022ZD0210300 | Zhengtuo Zhao |

The funders had no role in study design, data collection and interpretation, or the decision to submit the work for publication.

### Author contributions

Tian-Yi Wang, Conceptualization, Formal analysis, Investigation, Visualization, Writing - original draft, Project administration, Writing – review and editing, performed all the analyses on electrophysiology data and optogenetic manipulation, all the related behavior training experiments; Chengcong Feng, Conceptualization, Resources, Formal analysis, Investigation, Methodology, performed all the implantation surgeries, video analyses of mouse oral-facial movements during task learning and related behavior training, design of ultra-flexible microelectrode array devices; Chengyao Wang, Resources,

Methodology, design and fabrication of ultra-flexible microelectrode array devices; Chi Ren, Supervision, Funding acquisition, Writing – review and editing; Zhengtuo Zhao, Conceptualization, Supervision, Funding acquisition, Writing – review and editing

Author ORCIDs
Tian-Yi Wang ⓘ https://orcid.org/0000-0001-6488-339X
Chengcong Feng ⓘ https://orcid.org/0009-0001-5948-5722
Zhengtuo Zhao ⓘ https://orcid.org/0000-0003-2476-1560

Ethics
EthicsAll experimental procedures were approved by the Animal Care and Use Committee at the Center for Excellence in Brain Science and Intelligence Technology, Chinese Academy of Sciences, protocol # NA-056–2020.

Reviewer #1 (Public review): https://doi.org/10.7554/eLife.108083.3.sa1
Reviewer #2 (Public review): https://doi.org/10.7554/eLife.108083.3.sa2
Reviewer #3 (Public review): https://doi.org/10.7554/eLife.108083.3.sa3
Author response https://doi.org/10.7554/eLife.108083.3.sa4

# Additional files

## Supplementary files
MDAR checklist

## Data availability
Spiking data and behavior data analyzed during this study and scripts of main analyses are available on Dryad.

The following dataset was generated:

| Author(s) | Year | Dataset title | Dataset URL | Database and Identifier |
| --- | --- | --- | --- | --- |
| Tian-Yi W, Chengcong F, Chengyao W, Chi R, Zhengtuo Z | 2025 | Data from: Dynamics of mesoscale brain network during decision-making learning revealed by chronic, large-scale single-unit recording | https://doi.org/10.5061/dryad.cnp5hqcj2 | Dryad Digital Repository, 10.5061/dryad.cnp5hqcj2 |

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
