## [Editor Report · eLife Assessment]

This study presents experiments suggesting intriguing mesoscale reorganization of functional connectivity across distributed cortical and subcortical circuits during learning. The approach is technically impressive, and the results are potentially of **valuable** significance. The authors have also made a clear effort to address concerns in revision. However, the strength of evidence remains **incomplete**. Acquisition of data from additional animals in the primary experiment could bolster these findings.

---

## [Referee Report · Reviewer #1 (Public review)]

Summary:

This study aims to address an important and timely question: how does the mesoscale architecture of cortical and subcortical circuits reorganize during sensorimotor learning? By using high-density, chronically implanted ultra-flexible electrode arrays, the authors track spiking activity across ten brain regions as mice learn a visual Go/No-Go task. The results indicate that learning leads to more sequential and temporally compressed patterns of activity during correct rejection trials, alongside changes in functional connectivity ranks that reflect shifts in the relative influence of visual, frontal, and motor areas throughout learning. The emergence of a more task-focused subnetwork is accompanied by broader and faster propagation of stimulus information across recorded regions.

Strengths:

A clear strength of this work is its recording approach. The combination of stable, high-throughput multi-region recordings over extended periods represents a significant advance for capturing learning-related network dynamics at the mesoscale. The conceptual framework is well motivated, building on prior evidence that decision-relevant signals are widely distributed across the brain. The analysis approach, combining functional connectivity rankings with information encoding metrics is well motivated but needs refinement. These results provide some valuable evidence of how learning can refine both the temporal precision and the structure of interregional communication, offering new insights into circuit reconfiguration during learning.

Weaknesses:

Several important aspects of the evidence remain incomplete. In particular, it is unclear whether the reported changes in connectivity truly capture causal influences, as the rank metrics remain correlational and show discrepancies with the manipulation results. The absolute response onset latencies also appear slow for sensory-guided behavior in mice, and it is not clear whether this reflects the method used to define onset timing or factors such as task structure or internal state. Furthermore, the small number of animals, combined with extensive repeated measures, raises questions about statistical independence and how multiple comparisons were controlled. The optogenetic experiments, while intended to test the functional relevance of rank-increasing regions, leave it unclear how effectively the targeted circuits were silenced. Without direct evidence of reliable local inhibition, the behavioral effects or lack thereof are difficult to interpret.

---

## [Referee Report · Reviewer #2 (Public review)]

Summary:

Wang et al. measure from 10 cortical and subcortical brain as mice learn a go/no-go visual discrimination task. They found that during learning, there is a reshaping of inter-areal connections, in which a visual-frontal subnetwork emerges as mice gain expertise. Also visual stimuli decoding became more widespread post-learning. They also perform silencing experiments and find that OFC and V2M are important for the learning process. The conclusion is that learning evoked a brain-wide dynamic interplay between different brain areas that together may promote learning.

Strengths:

The manuscript is written well and the logic is rather clear. I found the study interesting and of interest to the field. The recording method is innovative and requires exceptional skills to perform. The outcomes of the study are significant, highlighting that learning evokes a widespread and dynamics modulation between different brain areas, in which specific task-related subnetworks emerge.

Weaknesses:

I had some major concerns that make the claims of the study less convincing: Low number of mice, insufficient movement analysis, figure visualization and analytic methods.

Nevertheless, I had several major concerns:

(1) The number of mice was small for the ephys recordings. Although the authors start with 7 mice in Figure 1, they then reduce to 5 in panel F. And in their main analysis they minimize their analysis 6/7 sessions from 3 mice only. I couldn't find a rationale for this reduction, but in the methods they do mention that 2 mice were used for fruitless training, which I found no mention in the results. Moreover, in the early case all of the analysis is from 118 CR trials taken from 3 mice. In general, this is a rather low number of mice and trial numbers. I think it is quite essential to add more mice.

(2) Movement analysis was not sufficient. Mice learning a go/no-go task establish a movement strategy that is developed throughout learning and is also biased towards Hit trials. There is an analysis of movement in Fig. S4 but this is rather superficial. I was not even sure that the 3 mice in Figure S4 are the same 3 mice in the main figure. There should be also an analysis of movement as a function of time to see differences. Also for Hits and FAs. I give some more details below. In general, most of the results can be explained by the fact that as mice gain expertise, they move more (also in CR during specific times) which leads to more activation in frontal cortex and more coordination with visual areas. More needs to be done in terms of analysis, or at least a mention of this in the text.

(3) Most of the figures are over-detailed and it is hard to understand the take home message. Although the text is written succinctly and rather short, the figures are mostly overwhelming, especially figures 4-7. For example, Figure 4 presents 24 brain plots! For rank input and output rank during early and late stim and response periods, for early and expert and their difference. All in the same colormap. No significance shown at all. The Δrank maps for all cases look essentially identical across conditions. The division into early and late time periods is not properly justified. But the main take home message is positive Δrank in OFC, V2M, V1 and negative Δrank in ThalMD and Str. In my opinion, one trio maps is enough, and the rest could be bumped to the Supp, if at all. In general, the figures in several cases do not convey the main take home messages.

(4) Analysis is sometimes not intuitive enough. For example, the rank analysis of input and output rank seemed a bit over complex. Figure 3 was hard to follow (although a lot of effort was made by the authors to make it clearer). Was there any difference between output and input analysis? Also time period seem sometimes redundant. Also, there are other network analysis that can be done which are a bit more intuitive. The use of rank within the 10 areas was not the most intuitive. Even a dimensionality reduction along with clustering can be used as an alternative. In my opinion, I don't think the authors should completely redo their analysis, but maybe mention the fact that other analyses exist.

Reviewer comments to the authors' revision:

Thank you for the extensive revision. Most of my concerns were answered and the manuscript is much improved. Still, there are some major issues that remain unconvincing:

(1) The number of learning mice is only 3 which is substantially low as compared to other studies in the field. Thus, statistics are across trials and session pooled from all mice. This is a big limitation in supporting the authors' claims

(2) There is no measurement of movement during the task. Since there are already several studies showing that movement has a strong effect on brain-wide dynamics, and since it is well known that mice change their body movement during learning (at least some mice) the authors cannot disentangle between learning-related and movement-related dynamics. This issue is properly discussed in the paper and also partially addressed with a control group where movement was measured without neural recordings.

(3) The authors do not know exactly where they recorded from, with emphasis on subcortical areas. The authors partially address this in a separate cohort where they regenerate the reproducibility rate of penetration locations, but still this is not a complete address to this concern.

Given the issues above, I strongly recommend including additional mice with body movement measurement in the future. Great job and congratulations on this study!

---

## [Referee Report · Reviewer #3 (Public review)]

Summary:

In the manuscript " Dynamics of mesoscale brain network during decision-making learning revealed by chronic, large-scale single-unit recording", Wang et al investigated mesoscale network reorganization during visual stimulus discrimination learning in mice using chronic, large-scale single-unit recordings across 10 cortical/subcortical regions. During learning, mice improved task performance mainly by suppressing licking on no-go trials. The authors found that learning induced restructuring of functional connectivity, with visual (V1, V2M) and frontal (OFC, M2) regions forming a task-relevant subnetwork during the acquisition of correct No-Go (CR) trials. Learning also compressed sequential neural activation and broadened stimulus encoding across regions. In addition, a region's network connectivity rank correlated with its timing of peak visual stimulus encoding. Optogenetic inhibition of orbitofrontal cortex (OFC) and high order visual cortex (V2M) impaired learning, validating its role in learning. The work highlights how mesoscale networks underwent dynamic structuring during learning.

Strengths:

The use of ultra-flexible microelectrode arrays (uFINE-M) for chronic, large-scale recordings across 10 cortical/subcortical regions in behaving mice represents a significant methodological advancement. The ability to track individual units over weeks across multiple brain areas will provide a rare opportunity to study mesoscale network plasticity.

While limited in scope, optogenetic inhibition of OFC and V2M directly ties connectivity rank changes to behavioral performance, adding causal depth to correlational observations.

Weaknesses:

The weakness is also related to the strength provided by the method. While the method in principle enables chronic tracking of individual units, the authors have not showed chronically tracked neurons across learning. Without demonstrating that and taking advantage of analyzing chronically tracked neurons, this approach is not different from acute recording in individual days across learning, weaking the attractiveness of the methodology and this study.

Another weakness is that major results are based on analyses of functional connectivity. Functional connection strengthen across areas is ranked 1-10 based on relative strength. And the regional input/out is compared across learning. This approach reveals differential changes in some cortical and subcortical areas. In my view, learning-related changes should be validated using complementary methods.

---

## [Author Response]

The following is the authors’ response to the original reviews.

**Public Reviews:**

**Reviewer #1 (Public review):**
Weaknesses:The technical approach is strong and the conceptual framing is compelling, but several aspects of the evidence remain incomplete. In particular, it is unclear whether the reported changes in connectivity truly capture causal influences, as the rank metrics remain correlational and show discrepancies with the manipulation results.

We agree that our functional connectivity ranking analyses cannot establish causal influences. As discussed in the manuscript, besides learning-related activity changes, the functional connectivity may also be influenced by neuromodulatory systems and internal state fluctuations. In addition, the spatial scope of our recordings is still limited compared to the full network implicated in visual discrimination learning, which may bias the ranking estimates. In future, we aim to achieve broader region coverage and integrate multiple complementary analyses to address the causal contribution of each region.

The absolute response onset latencies also appear slow for sensory-guided behavior in mice, and it is not clear whether this reflects the method used to define onset timing or factors such as task structure or internal state.

We believe this may be primarily due to our conservative definition of onset timing. Specifically, we required the firing rate to exceed baseline (t-test, p < 0.05) for at least 3 consecutive 25-ms time windows. This might lead to later estimates than other studies, such as using the latency to the first spike after visual stimulus onset (Siegle et al., 2021) or the time to half-max response (Goldbach, Akitake, Leedy, & Histed, 2021).

The estimation of response onset latency in our study may also be affected by potential internal state fluctuations of the mice. We used the time before visual stimulus onset as baseline firing, since firing rates in this period could be affected by trial history, we acknowledge this may increase the variability of the baseline, thus increase the difficulty to statistically detect the onset of response.

Still, we believe these concerns do not affect the observation of the formation of compressed activity sequence in CR trials during learning.

Furthermore, the small number of animals, combined with extensive repeated measures, raises questions about statistical independence and how multiple comparisons were controlled.

We agree that a larger sample size would strengthen the robustness of the findings. However, as noted above, the current dataset has inherent limitations in both the number of recorded regions and the behavioral paradigm. Given the considerable effort required to achieve sufficient unit yields across all targeted regions, we wish to adjust the set of recorded regions, improve behavioral task design, and implement better analyses in future studies. This will allow us to both increase the number of animals and extract more precise insights into mesoscale dynamics during learning.

The optogenetic experiments, while intended to test the functional relevance of rank increasing regions, leave it unclear how effectively the targeted circuits were silenced. Without direct evidence of reliable local inhibition, the behavioral effects or lack thereof are difficult to interpret.

We appreciate this important point. Due to the design of the flexible electrodes and the implantation procedure, bilateral co-implantation of both electrodes and optical fibers was challenging, which prevented us from directly validating the inhibition effect in the same animals used for behavior. In hindsight, we could have conducted parallel validations using conventional electrodes, and we will incorporate such controls in future work to provide direct evidence of manipulation efficacy.

Details on spike sorting are limited.

We have provided more details on spike sorting in method section, including the exact parameters used in the automated sorting algorithm and the subsequent manual curation criteria.

**Reviewer #2 (Public review):**
Weaknesses:I had several major concerns:(1) The number of mice was small for the ephys recordings. Although the authors start with 7 mice in Figure 1, they then reduce to 5 in panel F. And in their main analysis, they minimize their analysis to 6/7 sessions from 3 mice only. I couldn't find a rationale for this reduction, but in the methods they do mention that 2 mice were used for fruitless training, which I found no mention in the results. Moreover, in the early case, all of the analysis is from 118 CR trials taken from 3 mice. In general, this is a rather low number of mice and trial numbers. I think it is quite essential to add more mice.

We apologize for the confusion. As described in the Methods section, 7 mice (Figure 1B) were used for behavioral training without electrode array or optical fiber implants to establish learning curves, and an additional 5 mice underwent electrophysiological recordings (3 for visual-based decision-making learning and 2 for fruitless learning).

As we noted in our response to Reviewer #1, the current dataset has inherent limitations in both the number of recorded regions and the behavioral paradigm. Given the considerable effort required to achieve high-quality unit yields across all targeted regions, we wish to adjust the set of recorded regions, improve behavioral task design, and implement better analyses in future studies. These improvements will enable us to collect data from a larger sample size and extract more precise insights into mesoscale dynamics during learning.

(2) Movement analysis was not sufficient. Mice learning a go/no-go task establish a movement strategy that is developed throughout learning and is also biased towards Hit trials. There is an analysis of movement in Figure S4, but this is rather superficial. I was not even sure that the 3 mice in Figure S4 are the same 3 mice in the main figure. There should be also an analysis of movement as a function of time to see differences. Also for Hits and FAs. I give some more details below. In general, most of the results can be explained by the fact that as mice gain expertise, they move more (also in CR during specific times) which leads to more activation in frontal cortex and more coordination with visual areas. More needs to be done in terms of analysis, or at least a mention of this in the text.

Due to the limitation in the experimental design and implementation, movement tracking was not performed during the electrophysiological recordings, and the 3 mice shown in Figure S4 (now S5) were from a separate group. We have carefully examined the temporal profiles of mouse movements and found it did not fully match the rank dynamics for all regions, and we have added these results and related discussion in the revised manuscript. However, we acknowledge the observed motion energy pattern could explain some of the functional connection dynamics, such as the decrease in face and pupil motion energy could explain the reduction in ranks for striatum.

Without synchronized movement recordings in the main dataset, we cannot fully disentangle movement-related neural activity from task-related signals. We have made this limitation explicit in the revised manuscript and discuss it as a potential confound, along with possible approaches to address it in future work.

(3) Most of the figures are over-detailed, and it is hard to understand the take-home message. Although the text is written succinctly and rather short, the figures are mostly overwhelming, especially Figures 4-7. For example, Figure 4 presents 24 brain plots! For rank input and output rank during early and late stim and response periods, for early and expert and their difference. All in the same colormap. No significance shown at all. The Δrank maps for all cases look essentially identical across conditions. The division into early and late time periods is not properly justified. But the main take home message is positive Δrank in OFC, V2M, V1 and negative Δrank in ThalMD and Str. In my opinion, one trio map is enough, and the rest could be bumped to the Supplementary section, if at all. In general, the figure in several cases do not convey the main take home messages. See more details below.

We thank the reviewer for this valuable critique. The statistical significance corresponding to the brain plots (Figure 4 and Figure 5) was presented in Figure S3 and S5 (now Figure S5 and S7 in the revised manuscript), but we agree that the figure can be simplified to focus on the key results.

In the revised manuscript, we have condensed these figures to focus on the most important comparisons to make the visual presentation more concise and the take-home message clearer.

(4) The analysis is sometimes not intuitive enough. For example, the rank analysis of input and output rank seemed a bit over complex. Figure 3 was hard to follow (although a lot of effort was made by the authors to make it clearer). Was there any difference between the output and input analysis? Also, the time period seems redundant sometimes. Also, there are other network analysis that can be done which are a bit more intuitive. The use of rank within the 10 areas was not the most intuitive. Even a dimensionality reduction along with clustering can be used as an alternative. In my opinion, I don't think the authors should completely redo their analysis, but maybe mention the fact that other analyses exist

We appreciate the reviewer’s comment. In brief, the input- and output-rank analyses yielded largely similar patterns across regions in CR trials, although some differences were observed in certain areas (e.g., striatum) in Hit trials, where the magnitude of rank change was not identical between input and output measures. We have condensed the figures to only show averaged rank results, and the colormap was updated to better covey the message.

We did explore dimensionality reduction applied to the ranking data. However, the results were not intuitive as well and required additional interpretation, which did not bring more insights. Still, we acknowledge that other analysis approaches might provide complementary insights.

**Reviewer #3 (Public review):**
Weaknesses:The weakness is also related to the strength provided by the method. It is demonstrated in the original method that this approach in principle can track individual units for four months (Luan et al, 2017). The authors have not showed chronically tracked neurons across learning. Without demonstrating that and taking advantage of analyzing chronically tracked neurons, this approach is not different from acute recording across multiple days during learning. Many studies have achieved acute recording across learning using similar tasks. These studies have recorded units from a few brain areas or even across brain-wide areas.

We appreciate the reviewer’s important point. We did attempt to track the same neurons across learning in this project. However, due to the limited number of electrodes implanted in each brain region, the number of chronically tracked neurons in each region was insufficient to support statistically robust analyses. Concentrating probes in fewer regions would allow us to obtain enough units tracked across learning in future studies to fully exploit the advantages of this method.

Another weakness is that major results are based on analyses of functional connectivity that is calculated using the cross-correlation score of spiking activity (TSPE algorithm). Functional connection strengthen across areas is then ranked 1-10 based on relative strength. Without ground truth data, it is hard to judge the underlying caveats. I'd strongly advise the authors to use complementary methods to verify the functional connectivity and to evaluate the mesoscale change in subnetworks. Perhaps the authors can use one key information of anatomy, i.e. the cortex projects to the striatum, while the striatum does not directly affect other brain structures recorded in this manuscript

We agree that the functional connectivity measured in this study relies on statistical correlations rather than direct anatomical connections. We plan to test the functional connection data with shorter cross-correlation delay criteria to see whether the results are consistent with anatomical connections and whether the original findings still hold.

**Recommendations for the authors:**

**Reviewer #1 (Recommendations for the authors):**
(1) The small number of mice, each contributing many sessions, complicates the interpretation of the data. It is unclear how statistical analyses accounted for the small sample size, repeated measures, and non-independence across sessions, or whether multiple comparisons were adequately controlled.

We realized the limitation from the small number of animal subjects, yet the difficulty to achieve sufficient unit yields across all regions in the same animal restricted our sample size. Though we agree that a larger sample size would strengthen the robustness of the findings, however, as noted below the current dataset has inherent limitations in both the scope of recorded regions and the behavioral paradigm.

Given the considerable effort required to achieve sufficient unit yields across all targeted regions, we wish to adjust the set of recorded regions, improve behavioral task design, and implement better analyses in future studies. This will allow us to both increase the number of animals and extract more precise insights into mesoscale dynamics during learning.

(2) The ranking approach, although intuitive for visualizing relative changes in connectivity, is fundamentally descriptive and does not reflect the magnitude or reliability of the connections. Converting raw measures into ordinal ranks may obscure meaningful differences in strength and can inflate apparent effects when the underlying signal is weak.

We agree with this important point. As stated in the manuscript, our motivation in taking the ranking approach was that the differences in firing rates might bias cross-correlation between spike trains, making raw accounts of significant neuron pairs difficult to compare across conditions, but we acknowledge the ranking measures might obscure meaningful differences or inflate weak effects in the data.

We added the limitations of ranking approach in the discussion section and emphasized the necessity in future studies for better analysis approaches that could provide more accurate assessment of functional connection dynamics without bias from firing rates.

(3) The absolute response onset latencies also appear quite slow for sensory-guided behavior in mice, and it remains unclear whether this reflects the method used to determine onset timing or factors such as task design, sensorimotor demands, or internal state. The approach for estimating onset latency by comparing firing rates in short windows to baseline using a t-test raises concerns about robustness, as it may be sensitive to trial-to-trial variability and yield spurious detections.

We agree this may be primarily due to our conservative definition of onset timing. Specifically, we required the firing rate to exceed baseline (t-test, p < 0.05) for at least 3 consecutive 25-ms time windows. This might lead to later estimates than other studies, such as using the latency to the first spike after visual stimulus onset (Siegle et al., 2021) or the time to half-max response (Goldbach, Akitake, Leedy, & Histed, 2021).

The estimation of response onset latency in our study may also be affected by potential internal state fluctuations of the mice. We used the time before visual stimulus onset as baseline firing, since firing rates in this period could be affected by trial history, we acknowledge this may increase the variability of the baseline, thus increase the difficulty to statistically detect the onset of response.

Still, we believe these concerns do not affect the observation of the formation of compressed activity sequence in CR trials during learning.

(4) Details on spike sorting are very limited. For example, defining single units only by an interspike interval threshold above one millisecond may not sufficiently rule out contamination or overlapping clusters. How exactly were neurons tracked across days (Figure 7B)?

We have added more details on spike sorting, including the processing steps and important parameters used in the automated sorting algorithm. Only the clusters well isolated in feature space were accepted in manual curation.

We attempted to track the same neurons across learning in this project. However, due to the limited number of electrodes implanted in each brain region, the number of chronically tracked neurons in each region was insufficient to support statistically robust analyses.

This is now stated more clearly in the discussion section.

(5) The optogenetic experiments, while designed to test the functional relevance of rank-increasing regions, also raise questions. The physiological impact of the inhibition is not characterized, making it unclear how effectively the targeted circuits were actually silenced. Without clearer evidence that the manipulations reliably altered local activity, the interpretation of the observed or absent behavioral effects remains uncertain.

We appreciate this important point. Due to the design of the flexible electrodes and the implantation procedure, bilateral co-implantation of both electrodes and optical fibers was challenging, which prevented us from directly validating the inhibition effect in the same animals used for behavior. In hindsight, we could have conducted parallel validations using conventional electrodes, and we will incorporate such controls in future work to provide direct evidence of manipulation efficacy.

(6) The task itself is relatively simple, and the anatomical coverage does not include midbrain or cerebellar regions, limiting how broadly the findings can be generalized to more flexible or ethologically relevant forms of decision-making.

We appreciate this advice and have expanded the existing discussion to more explicitly state that the relatively simple task design and anatomical coverage might limit the generalizability of our findings.

(7) The abstract would benefit from more consistent use of tense, as the current mix of past and present can make the main findings harder to follow. In addition, terms like "mesoscale network," "subnetwork," and "functional motif" are used interchangeably in places; adopting clearer, consistent terminology would improve readability.

We have changed several verbs in abstract to past form, and we now adopted a more consistent terminology by substituting “functional motif” as “subnetwork”. We still feel the use of

“mesoscale network” and “subnetwork” could emphasize different aspects of the results according to the context, so these words are kept the same.

(8) The discussion could better acknowledge that the observed network changes may not reflect task-specific learning alone but could also arise from broader shifts in arousal, attention, or motivation over repeated sessions.

We have expanded the existing discussion to better acknowledge the possible effects from broader shifts in arousal, attention, or motivation over repeated sessions.

(9) The figures would also benefit from clearer presentation, as several are dense and not straightforward to interpret. For example, Figure S8 could be organized more clearly to highlight the key comparisons and main message

We have simplified the over-detailed brain plots in Figure 4-5, and the plots in Figure 6 and S8 (now S10 in the revised manuscript).

(10) Finally, while the manuscript notes that data and code are available upon request, it would strengthen the study's transparency and reproducibility to provide open access through a public repository, in line with best practices in the field.

The spiking data, behavior data and codes for the core analyses in the manuscript are now shared in pubic repository (Dryad). And we have changed the description in the Data Availability secition accordingly.

**Reviewer #2 (Recommendations for the authors):**
(A) Introduction:(1) "Previous studies have implicated multiple cortical and subcortical regions in visual task learning and decision-making". No references here, and also in the next sentence.

The references were in the following introduction and we have added those references here as well.

We also added one review on cortical-subcortical neural correlates in goal-directed behavior (Cruz et al., 2023).

(2) Intro: In general, the citation of previous literature is rather minimal, too minimal. There is a lot of studies using large scale recordings during learning, not necessarily visual tasks. An example for brain-wide learning study in subcortical areas is Sych et al. 2022 (cell reports). And for wide-field imaging there are several papers from the Helmchen lab and Komiyama labs, also for multi-area cortical imaging.

We appreciate this advice. We included mainly visual task learning literature to keep a more focused scope around the regions and task we actually explored in this study. We fear if we expand the intro to include all the large-scale imaging/recording studies in learning field, the background part might become too broad.

We have included (Sych, Fomins, Novelli, & Helmchen, 2022) for its relevance and importance in the field.

(3) In the intro, there is only a mention of a recording of 10 brain regions, with no mention of which areas, along with their relevance to learning. This is mentioned in the results, but it will be good in the intro.

The area names are now added in intro.

(B) Results:(1) Were you able to track the same neurons across the learning profile? This is not stated clearly.

We did attempt to track the same neurons across learning in this project. However, due to the limited number of electrodes implanted in each brain region, the number of chronically tracked neurons in each region was insufficient to support statistically robust analyses.

We now stated this more clearly in the discussion section.

(2) Figure 1 starts with 7 mice, but only 5 mice are in the last panel. Later it goes down to 3 mice. This should be explained in the results and justified.

We apologize for the confusion. As described in the Methods section, 7 mice (Figure 1B) were used for behavioral training without electrode array or optical fiber implants to establish learning curves, and an additional 5 mice underwent electrophysiological recordings (3 for visual-based decision-making learning and 2 for fruitless learning).

(3) I can't see the electrode tracks in Figure 1d. If they are flexible, how can you make sure they did not bend during insertion? I couldn't find a description of this in the methods also.

The electrode shanks were ultra-thin (1-1.5 µm) and it was usually difficult to recover observable tracks or electrodes in section.

The ultra-flexible probes could not penetrate brain on their own (since they are flexible), and had to be shuttled to position by tungsten wires through holes designed at the tip of array shanks. The tungsten wires were assembled to the electrode array before implantation; this was described in the section of electrode array fabrication and assembly. We also included the description about the retraction of the guiding tungsten wires in the surgery section to avoid confusion.

As an further attempt to verify the accuracy of implantation depth, we also measured the repeatability of implantation in a group of mice and found a tendency for the arrays to end in slightly deeper location in cortex (142.1 ± 55.2 μm, n = 7 shanks), and slightly shallower location in subcortical structure (-122.6 ± 71.7 μm, n = 7 shanks). We added these results as new Figure S1 to accompany Figure 1.

(4) In the spike rater in 1E, there seems to be ~20 cells in V2L, for example, but in 1F, the number of neurons doesn't go below 40. What is the difference here?

We checked Figure 1F, the plotted dots do go below 40 to ~20. Perhaps the file that reviewer received wasn’t showing correctly?

(5) The authors focus mainly on CR, but during learning, the number of CR trials is rather low (because they are not experts). This can also be seen in the noisier traces in Figure 2a. Do the authors account for that (for example by taking equal trials from each group)?

We accounted this by reconstructing bootstrap-resampled datasets with only 5 trials for each session in both the early stage and the expert stage. The mean trace of the 500 datasets again showed overall decrease in CR trial firing rate during task learning, with highly similar temporal dynamics to the original data.

The figure is now added to supplementary materials (as Figure S3 in the revised manuscript).

(6) From Figure 2a, it is evident that Hit trials increase response when mice become experts in all brain areas. The authors have decided to focus on the response onset differences in CRs, but the Hit responses display a strong difference between naïve and expert cases.

Judged from the learning curve in this task the mice learned to inhibit its licking action when the No-Go stimuli appeared, which is the main reason we focused on these types of trials.

The movement effects and potential licking artefacts in Hit trials also restricted our interpretation of these trials.

(7) Figure 3 is still a bit cumbersome. I wasn't 100% convinced of why there is a need to rank the connection matrix. I mean when you convert to rank, essentially there could be a meaningful general reduction in correlation, for example during licking, and this will be invisible in the ranking system. Maybe show in the supp non-ranked data, or clarify this somehow

We agree with this important point. As stated in the manuscript and response to Reviewer #1, our motivation in taking the ranking approach was that the differences in firing rates could bias cross-correlation between spike trains, making raw accounts of significant neuron pairs difficult to compare across conditions, but we acknowledge the ranking measures might obscure meaningful differences or inflate weak effects in the data.

We added the limitations of ranking approach in the discussion section and emphasized the necessity in future studies for better analysis approaches that could provide more accurate assessment of functional connection dynamics without bias from firing rates.

(8) Figure 4a x label is in manuscript, which is different than previous time labels, which were seconds.

We now changed all time labels from Figure 2 to milliseconds.

(9) Figure 4 input and output rank look essentially the same.

We have compressed the brain plots in Figures 4-5 to better convey the take-home message.

(10) Also, what is the late and early stim period? Can you mark each period in panel A? Early stim period is confusing with early CR period. Same for early respons and late response.

The definition of time periods was in figure legends. We now mark each period out to avoid confusion.

(11) Looking at panel B, I don't see any differences between delta-rank in early stim, late stim, early response, and late response. Same for panel c and output plots.

The rankings were indeed relatively stable across time periods. The plots are now compressed and showed a mean rank value.

(12) Panels B and C are just overwhelming and hard to grasp. Colors are similar both to regular rank values and delta-rank. I don't see any differences between all conditions (in general). In the text, the authors report only M2 to have an increase in rank during the response period. Late or early response? The figure does not go well with the text. Consider minimizing this plot and moving stuff to supplementary.

The colormap are now changed to avoid confusion, and brain plots are now compressed.

(13) In terms of a statistical test for Figure 4, a two-way ANOVA was done, but over what? What are the statistics and p-values for the test? Is there a main effect of time also? Is their a significant interaction? Was this done on all mice together? How many mice? If I understand correctly, the post-hoc statistics are presented in the supplementary, but from the main figure, you cannot know what is significant and what is not.

For these figures we were mainly concerned with the post-hoc statistics which described the changes in the rankings of each region across learning.

We have changed the description to “t-test with Sidak correction” to avoid the confusion.

(14) In the legend of Figure 4, it is reported that 610 expert CR trials from 6 sessions, instead of 7 sessions. Why was that? Also, like the previous point, why only 3 mice?

Behavior data of all the sessions used were shown in Figure S1. There were only 3 mice used for the learning group, the difficulty to achieve sufficient unit yields across all regions in the same animal restricted our sample size

(15) Body movement analysis: was this done in a different cohort of mice? Only now do I understand why there was a division into early and late stim periods. In supp 4, there should be a trace of each body part in CR expert versus naïve. This should also be done for Hit trials as a sanity check. I am not sure that the brightness difference between consecutive frames is the best measure. Rather try to calculate frame-to frame correlation. In general, body movement analysis is super important and should be carefully analyzed.

Due to the limitation in the experimental design and implementation, movement tracking was not performed during the electrophysiological recordings, and the 3 mice shown in Figure S4 (now S5) were from a separate group. We have carefully examined the temporal profiles of mouse movements and found it did not fully match the rank dynamics for all regions, and we have added these results and related discussion in the revised manuscript. However, we acknowledge the observed motion energy pattern could explain some of the functional connection dynamics, such as the decrease in face and pupil motion energy could explain the reduction in ranks for striatum.

Without synchronized movement recordings in the main dataset, we cannot fully disentangle movement-related neural activity from task-related signals. We have made this limitation explicit in the revised manuscript and discuss it as a potential confound, along with possible approaches to address it in future work.

(16) For Hit trials, in the striatum, there is an increase in input rank around the response period, and from Figure S6 it is clear that this is lick-related. Other than that, the authors report other significant changes across learning and point out to Figure 5b,c. I couldn't see which areas and when it occurred.

We did naturally expect the activity in striatum to be strongly related to movement.

With Figure S6 (now S7) we wished to show that the observed rank increase for striatum could not simply be attributed to changes in time of lick initiation.

As some readers may argue that during learning the mice might have learned to only intensely lick after response signal onset, causing the observed rise of input rank after response signal, we realigned the spikes in each trial to the time of the first lick, and a strong difference could still be observed between early training stage and expert training stage.

We still cannot fully rule out the effects from more subtle movement changes, as the face motion energy did increase in early response period. This result and related discussion has been added to the results section of revised manuscript.

(17) Figure 6, again, is rather hard to grasp. There are 16 panels, spread over 4 areas, input and output, stim and response. What is the take home message of all this? Visually, it's hard to differentiate between each panel. For me, it seems like all the panels indicate that for all 4 areas, both in output and input, frontal areas increase in rank. This take-home message can be visually conveyed in much less tedious ways. This simpler approach is actually conveyed better in the text than in the figures themselves. Also, the whole explanation on how this analysis was done, was not clear from the text. If I understand it, you just divided and ranked the general input (or output) into individual connections? If so, then this should be better explained.

We appreciate this advice and we have compressed the figures to better convey the main message.The rankings for Figure 6 and Figure S8 (now Figure S9) was explained in the left panel of Figure 3C. Each non-zero element in the connection matrix was ranked to value from 1-10, with a value of 10 represented the 10% strongest non-zero elements in the matrix.

We have updated the figure legends of Figure 3, and we have also updated the description in methods (Connection rank analyses) to give a clearer description of how the analyses were applied in subsequent figures.

(18) Figure 7: Here, the authors perform a ROC analysis between go and no-go stimuli. They balance between choice, but there is still an essential difference between a hit and a FA in terms of movement and licks. That is maybe why there is a big difference in selective units during the response period. For example, during a Hit trial the mouse licks and gets a reward, resulting in more licking and excitement. In FAs,the mouse licks, but gets punished, which causes a reduction in additional licking and movements. This could be a simple explanation why the ROC was good in the late response period. Body movement analysis of Hit and FA should be done as in Figure S4.

We appreciate this insightful advice.

Though we balanced the numbers of basic trial types, we couldn’t rule out the difference in the intrinsic movement amount difference in FA trials and Hit trials, which is likely the reason of large proportion of encoding neurons in response period.

We have added this discussion both in result section and discussion section along with the necessity of more carefully designed behavior paradigm to disentangle task information.

(19) The authors also find selective neurons before stimulus onset, and refer to trial history effects. This can be directly checked, that is if neurons decode trial history.

We attempted encoding analyses on trial history, but regrettably for our dataset we could not find enough trials to construct a dataset with fully balanced trial history, visual stimulus and behavior choice.

(20) Figure 7e. What is the interpretation for these results? That areas which peaked earlier had more input and output with other areas? So, these areas are initiating hubs? Would be nice to see ACC vs Str traces from B superimposed on each other. Having said this, the Str is the only area to show significant differences in the early stim period. But is also has the latest peak time. This is a bit of a discrepancy.

We appreciate this important point.

The limitation in the anatomical coverage of brain regions restricted our interpretation about these findings. They could be initiating hubs or earlier receiver of the true initiating hubs that were not monitored in our study.

The Str trace was in fact above the ACC trace, especially in the response period. This could be explained by the above advice 18: since we couldn’t rule out the difference in the intrinsic movement amount difference in FA trials and Hit trials, and considering striatum activity is strongly related to movement, the Str trace may reflect more in the motion related spike count difference between FA trials and Hit trials, instead of visual stimulus related difference.

This further shows the necessity of more carefully designed behavior paradigm to disentangle task information.

The striatum trace also in fact didn’t show a true double peak form as traces in other regions, it ramped up in the stimulus region and only peaked in response period. This description is now added to the results section.

In the early stim period, the Striatum did show significant differences in average percent of encoding neurons, as the encoding neurons were stably high in expert stage. The striatum activity is more directly affected Still the percentage of neurons only reached peak in late stimulus period.

(21) For the optogenetic silencing experiments, how many mice were trained for each group? This is not mentioned in the results section but only in the legend of Figure 8. This part is rather convincing in terms of the necessity for OFC and V2M

We have included the mice numbers in results section as well.

(C) Discussion(1) There are several studies linking sensory areas to frontal networks that should be mentioned, for example, Esmaeili et a,l 2022, Matteucci et al., 2022, Guo et a,l 2014,Gallero Salas et al, 2021, Jerry Chen et al, 2015. Sonja Hofer papers, maybe. Probably more.

We appreciate this advice. We have now included one of the mentioned papers (Esmaeili et al., 2022) in the results section and discussion section for its direct characterization of the enhanced coupling between somatosensory region and frontal (motor) region during sensory learning.The other studies mentioned here seem to focus more on the differences in encoding properties between regions along specific cortical pathways, rather than functional connection or interregional activity correlation, and we feel they are not directly related to the observations discussed.

(2) The reposted reorganization of brain-wide networks with shifts in time is best described also in Sych et al. 2021.

We regret we didn’t include this important research and we have now cited this in discussion section.

(3) Regarding the discussion about more widespread stimulus encoding after learning, the results indicate that the striatum emerges first in decoding abilities (Figure 7c left panel), but this is not discussed at all.

We briefly discussed this in the result section. We tend to attribute this to trial history signal in striatum, but since the structure of our data could not support a direct encoding analysis on trial history, we felt it might be inappropriate to over-interpret the results.

(4) An important issue which is not discussed is the contribution of movement which was shown to have a strong effect on brain-wide dynamics (Steinmetz et al 2019; Musall et al 2019; Stringer et al 2019; Gilad et al 2018) The authors do have some movement analysis, but this is not enough. At least a discussion of the possible effects of movement on learning-related dynamics should be added.

We have included these studies in discussion section accordingly. Since the movement analyses were done in a separate cohort of mice, we have made our limitation explicit in the revised manuscript and discuss it as a potential confound, along with possible approaches to address it in future work.

(D) Methods(1) How was the light delivery of the optogenetic experiments done? Via fiber implantation in the OFC? And for V2M? If the red laser was on the skull, how did it get to the OFC?

The fibers were placed on cortex surface for V2M group, and were implanted above OFC for OFC manipulation group. These were described in the viral injection part of the methods section.

(2) No data given on how electrode tracking was done post hoc

As noted in our response to the advice 3 in results section, the electrode shanks were ultra-thin (1-1.5 µm) and it was usually difficult to recover observable tracks or electrodes in section.

As an attempt to verify the accuracy of implantation depth, we measured the repeatability of implantation in a group of mice and found a tendency for the arrays to end in slightly deeper location in cortex (142.1 ± 55.2 μm, n = 7 shanks), and slightly shallower location in subcortical structure (-122.6 ± 71.7 μm, n = 7 shanks). We added these results as new Figure S1 to accompany Figure 1.

**Reviewer #3 (Recommendations for the authors):**
(1) The manuscript uses decision-making in the title, abstract and introduction. However, nothing is related to decision learning in the results section. Mice simply learned to suppress licking in no-go trials. This type of task is typically used to study behavioral inhibition. And consistent with this, the authors mainly identified changes related to network on no-go trials. I really think the title and main message is misleading. It is better to rephrase it as visual discrimination learning. In the introduction, the authors also reviewed multiple related studies that are based on learning of visual discrimination tasks.

We do view the Go/No-Go task as a specific genre of decision-making task, as there were literature that discussed this task as decision-making task under the framework of signal detection theory or updating of item values (Carandini & Churchland, 2013; Veling, Becker, Liu, Quandt, & Holland, 2022).

We do acknowledge the essential differences between the Go/No-Go task and the tasks that require the animal to choose between alternatives, and since we have now realized some readers may not accept this task as a decision task, we have changed the title to visual discrimination task as advised.

(2) Learning induced a faster onset on CR trials. As the no-go stimulus was not presented to mice during early stages of training, this change might reflect the perceptual learning of relevant visual stimulus after repeated presentation. This further confirms my speculation, and the decision-making used in the title is misleading.

We have changed the title to visual discrimination task accordingly.

(3) Figure 1E, show one hit trial. If the second 'no-go stimulus' is correct, that trial might be a false alarm trial as mice licked briefly. I'd like to see whether continuous licking can cause motion artifacts in recording.

We appreciate this important point. There were indeed licking artifacts with continuous licking in Hit trials, which was part of the reason we focused our analyses on CR trials. Opto-based lick detectors may help to reduce the artefacts in future studies.

(4) What is the rationale for using a threshold of d' < 2 as the early-stage data and d'>3 as expert stage data?

The thresholds were chosen as a result from trade-off based on practical needs to gather enough CR trials in early training stage, while maintaining a relatively low performance.

Assume the mice showed lick response in 95% of Go stimulus trials, then d' < 2 corresponded to the performance level at which the mouse correctly rejected less than 63.9% of No-Go stimulus trials, and d' > 3 corresponded to the performance level at which the mouse correctly rejected more than 91.2% of No-Go stimulus trials.

(5) Figure 2A, there is a change in baseline firing rates in V2M, MDTh, and Str. There is no discussion. But what can cause this change? Recording instability, problem in spiking sorting, or learning?

It’s highly possible that the firing rates before visual stimulus onset is affected by previous reward history and task engagement states of the mice. Notably, though recorded simultaneously in same sessions, the changes in CR trials baseline firing rates in the V2M region were not observed in Hit trials.

Thus, though we cannot completely rule out the possibility in recording instability, we see this as evidence of the effects on firing rates from changes in trial history or task engagement during learning.

References:

Carandini, M., & Churchland, A. K. (2013). Probing perceptual decisions in rodents. Nat Neurosci, 16(7), 824-831. doi:10.1038/nn.3410.

Cruz, K. G., Leow, Y. N., Le, N. M., Adam, E., Huda, R., & Sur, M. (2023).Cortical-subcortical interactions in goal-directed behavior. Physiol Rev, 103(1), 347-389. doi:10.1152/physrev.00048.2021

Esmaeili, V., Oryshchuk, A., Asri, R., Tamura, K., Foustoukos, G., Liu, Y., Guiet, R., Crochet, S., & Petersen, C. C. H. (2022). Learning-related congruent and incongruent changes of excitation and inhibition in distinct cortical areas. PLOS Biology, 20(5), e3001667. doi:10.1371/journal.pbio.3001667

Goldbach, H. C., Akitake, B., Leedy, C. E., & Histed, M. H. (2021). Performance in even a simple perceptual task depends on mouse secondary visual areas. Elife, 10, e62156. doi:10.7554/eLife.62156.

Siegle, J. H., Jia, X., Durand, S., Gale, S., Bennett, C., Graddis, N., Heller, G.,Ramirez, T. K., Choi, H., Luviano, J. A., Groblewski, P. A., Ahmed, R., Arkhipov, A., Bernard, A., Billeh, Y. N., Brown, D., Buice, M. A., Cain, N.,Caldejon, S., Casal, L., Cho, A., Chvilicek, M., Cox, T. C., Dai, K., Denman, D.J., de Vries, S. E. J., Dietzman, R., Esposito, L., Farrell, C., Feng, D., Galbraith, J., Garrett, M., Gelfand, E. C., Hancock, N., Harris, J. A., Howard, R., Hu, B.,Hytnen, R., Iyer, R., Jessett, E., Johnson, K., Kato, I., Kiggins, J., Lambert, S., Lecoq, J., Ledochowitsch, P., Lee, J. H., Leon, A., Li, Y., Liang, E., Long, F., Mace, K., Melchior, J., Millman, D., Mollenkopf, T., Nayan, C., Ng, L., Ngo, K., Nguyen, T., Nicovich, P. R., North, K., Ocker, G. K., Ollerenshaw, D., Oliver, M., Pachitariu, M., Perkins, J., Reding, M., Reid, D., Robertson, M., Ronellenfitch, K., Seid, S., Slaughterbeck, C., Stoecklin, M., Sullivan, D., Sutton, B., Swapp, J., Thompson, C., Turner, K., Wakeman, W., Whitesell, J. D., Williams, D., Williford, A., Young, R., Zeng, H., Naylor, S., Phillips, J. W., Reid, R. C., Mihalas, S., Olsen, S. R., & Koch, C. (2021). Survey of spiking in the mouse visual system reveals functional hierarchy. Nature, 592(7852), 86-92. doi:10.1038/s41586-020-03171-x

Sych, Y., Fomins, A., Novelli, L., & Helmchen, F. (2022). Dynamic reorganization of the cortico-basal ganglia-thalamo-cortical network during task learning. Cell Rep, 40(12), 111394. doi:10.1016/j.celrep.2022.111394

Veling, H., Becker, D., Liu, H., Quandt, J., & Holland, R. W. (2022). How go/no-go training changes behavior: A value-based decision-making perspective. Current Opinion in Behavioral Sciences, 47,101206.

doi:https://doi.org/10.1016/j.cobeha.2022.101206.